# De novo discovery of conserved gene clusters in microbial genomes with Spacedust

Ruoshi Zhang ⬤ [1], Milot Mirdita[2] & Johannes Söding ⬤ [1,3] ✉

Metagenomics has revolutionized environmental and human-associated microbiome studies. However, the limited fraction of proteins with known biological processes and molecular functions presents a major bottleneck. In prokaryotes and viruses, evolution favors keeping genes participating in the same biological processes colocalized as conserved gene clusters. Conversely, conservation of gene neighborhood indicates functional association. Here we present Spacedust, a tool for systematic, de novo discovery of conserved gene clusters. To find homologous protein matches, Spacedust uses fast and sensitive structure comparison with Foldseek. Partially conserved clusters are detected using novel clustering and order conservation *P* values. We demonstrate Spacedust's sensitivity with an all-versus-all analysis of 1,308 bacterial genomes, identifying 72,843 conserved gene clusters containing 58% of the 4.2 million genes. It recovered 95% of antiviral defense system clusters annotated by the specialized tool PADLOC. Spacedust's high sensitivity and speed will facilitate the annotation of large numbers of sequenced bacterial, archaeal and viral genomes.

In the past decade, metagenomics has accelerated the pace of research into microbial ecology and human-associated microbiomes and their intimate association with human health[1]. Hundreds of thousands of microbial and viral genomes assembled from shotgun metagenomics permit the study of microorganisms and their interactions with each other and their environment[2–4]. However, our ability to extract useful insights from such data is severely limited by the lack of functional information[5]. Even in well-studied ecosystems such as the human gut, for around 40% of genes neither molecular function nor biological process is annotatable[2].

The standard approach for protein function annotation is by homology inference, that is, by sequence similarity search to find the best match in reference databases such as InterPro, KEGG orthologs, COGs or SEED[6–8], and transferring the annotation if certain criteria are met[9–13]. Earlier approaches relied on sequence–sequence search tools such as BLAST. However, function can remain conserved even at sequence identities much below 20%, which these approaches cannot detect[14]. Therefore, modern approaches with increased sensitivity search with the query protein sequences through databases of

profile hidden Markov models (HMMs) or sequence profiles. These are precomputed from multiple sequence alignments of protein family members with the same or similar functions. Many databases of orthologous families have been developed to automate the process of clustering orthologous protein sequences together[7,15]. This approach is motivated by the 'orthology conjecture', which states that orthologous sequences are more likely to be functionally related than paralogous ones, although the difference appears to actually be small[16,17].

Integration of genomic context can improve the precision of ortholog clustering and functional annotation. Proteins do not work in isolation but cooperate with others in biological pathways. Evolution has a tendency to keep functionally associated genes closely together in prokaryotic and viral genomes. This can be a consequence of coexpressed genes sharing regulatory sequences or even forming part of the same transcription unit, an operon[18]. Clustering also maximizes the chances of horizontal transfer of useful gene modules, and it minimizes disruptions of functionally associated genes by genomic recombination[19–21]. Some methods exploit gene neighborhood conservation to increase specificity for identifying orthologs[22–25].

[1]Quantitative and Computational Biology, Max Planck Institute for Multidisciplinary Sciences, Göttingen, Germany. [2]School of Biological Sciences, Seoul National University, Seoul, Republic of Korea. [3]Campus Institute Data Science (CIDAS), University of Göttingen, Göttingen, Germany. ✉e-mail: soeding@mpinat.mpg.de

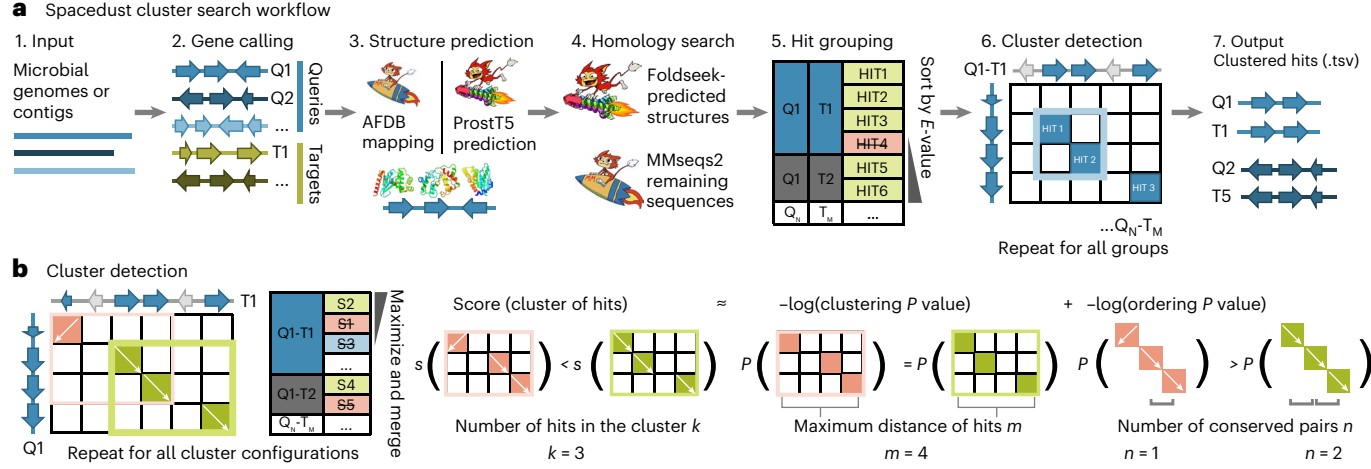

**Fig. 1 | Spacedust algorithm. a**, Workflow. **b**, Starting from single-gene clusters, clusters are iteratively merged if this increases their conservation score until all scores are maximal. Colored boxes indicate pairs of homologous proteins found by Foldseek. Cluster conservation is measured by combining a clustering *P* value and an ordering *P* value.

Others used the 'guilt by association' principle to find functionally associated genes (for example, see refs. 26,27).

Many methods have been designed to detect a specific type of cluster, such as biosynthetic gene clusters (BGCs)[28–31], phage defense systems[32–34], virulence and antibiotic resistance factors[35,36] or xenobiotic degradation pathways[37]. Most search the protein sequences from the query genome against a pre-assembled database of profile HMMs representing protein families typically occurring in these clusters. They then apply heuristic rules for what constitutes a valid cluster match.

A few tools aim to find genomic neighborhoods similar to a query neighborhood[38–41]. The sensitivity of these de novo cluster detection methods is limited severely by the use of sequence–sequence comparison tools such as BLAST or DIAMOND[42,43], compounding their ability to detect all but closely related conserved clusters[44]. They also do not scale up to more than a few hundred genomes in all-versus-all search mode, and some require strict conservation of gene order (colinearity). Some approaches find conserved clusters by first searching each of the genomes to be analyzed against a profile HMM database of orthologous groups, and then find clusters of genes with a similar composition of orthologs[45,46]. While this type of approach has improved sensitivity over the first, it requires a reference database of orthologous groups and, therefore, excludes the many proteins from as-yet-unknown families[47].

Spacedust is a tool for systematic de novo discovery of conserved gene clusters across multiple genomes. It finds all gene clusters significantly conserved between any two genomes in a set of input genomes. Conserved clusters are found by maximizing the statistical significance measured with two novel statistics assessing the degree of clustering and the degree of order and strand conservation. Because remote homologies are critical to achieve high sensitivity for detecting conserved gene clusters, Spacedust performs its homology searches with our new structure-based search tool Foldseek. Foldseek has similar sensitivity as the best structural comparison tools and much higher than sequence–sequence, sequence–profile and profile HMM searches[48,49].

Spacedust improves upon previous methods in several ways: (i) It is reference-free and can discover conserved clusters of any type and composition; (ii) its structure-based search maximizes its sensitivity for finding remotely related conserved gene clusters; (iii) its high speed allows for analyzing a large number of genomes for conserved gene clusters using all-versus-all searches; (iv) it integrates functional annotation of proteins to facilitate inference of function from cluster members; and (v) it offers a user-friendly Google Colab notebook.

We demonstrate the utility of Spacedust by detecting conserved clusters in an all-versus-all comparison of 1,308 representative bacterial genomes from different genera with a total of 4.2 million protein-coding genes. Spacedust recovers previously annotated gene clusters, for example, operons, antiviral defense systems and BGCs. It is able to assign 58% of all 4.2 million genes and 35% of genes without any annotation to conserved gene clusters. Spacedust also discovers the vast majority of antiphage defense systems in this dataset and achieves better results in identifying 207 manually annotated BGCs than three specialized tools.

## Results

### Spacedust algorithm

Spacedust takes as input a set *Q* of query genomes and a set *T* of target genomes (which may be equal to *Q*) and, for each pair $(q, t) \in Q \times T$ of query and target genome, it finds all gene clusters whose gene arrangement is at least partially conserved between *q* and *t* (Fig. 1 and Methods). For that purpose, Spacedust first identifies homologous matches ('hits') between proteins in *Q* and proteins in *T* using our sensitive structure search tool Foldseek[48] and our sequence search tool MMseqs2 (ref. 50; steps 1–5 in Fig. 1a). For every Q–T pair, it detects clusters of hits with significant conservation of gene neighborhood, using a greedy cluster detection algorithm (step 6 in Fig. 1a,b). The cluster detection starts with each protein hit in its own cluster and adds protein hits to the cluster matches one at a time. If the significance score of the cluster match improves, the addition is accepted and the algorithm continues, until the significance of the cluster matches cannot be improved further. The significance score is calculated as the sum of the negative logarithms of a clustering *P* value and an ordering *P* value. The clustering *P* value is the probability of finding 'by chance' at least *k* matches within a window of at most *m* genes in both the query and the target genome. The ordering *P* value is the probability to find 'by chance' at least *n* pairs of genes of the cluster match in conserved order in both genomes. The cluster detection algorithm thereby identifies positionally conserved clusters between all *Q–T* pairs of genomes. Optionally, the cluster matches for each query genome, aggregated across multiple reference genomes, can be visualized as a measure of conservation strength.

### A reference set of remotely conserved bacterial gene clusters

Despite the availability of tens of thousands of complete bacterial genomes, the gene cluster conservation landscape has yet to be surveyed systematically. To address this gap, we curated a dataset of 1,308 bacterial reference genomes, covering a broad phylogenetic range.

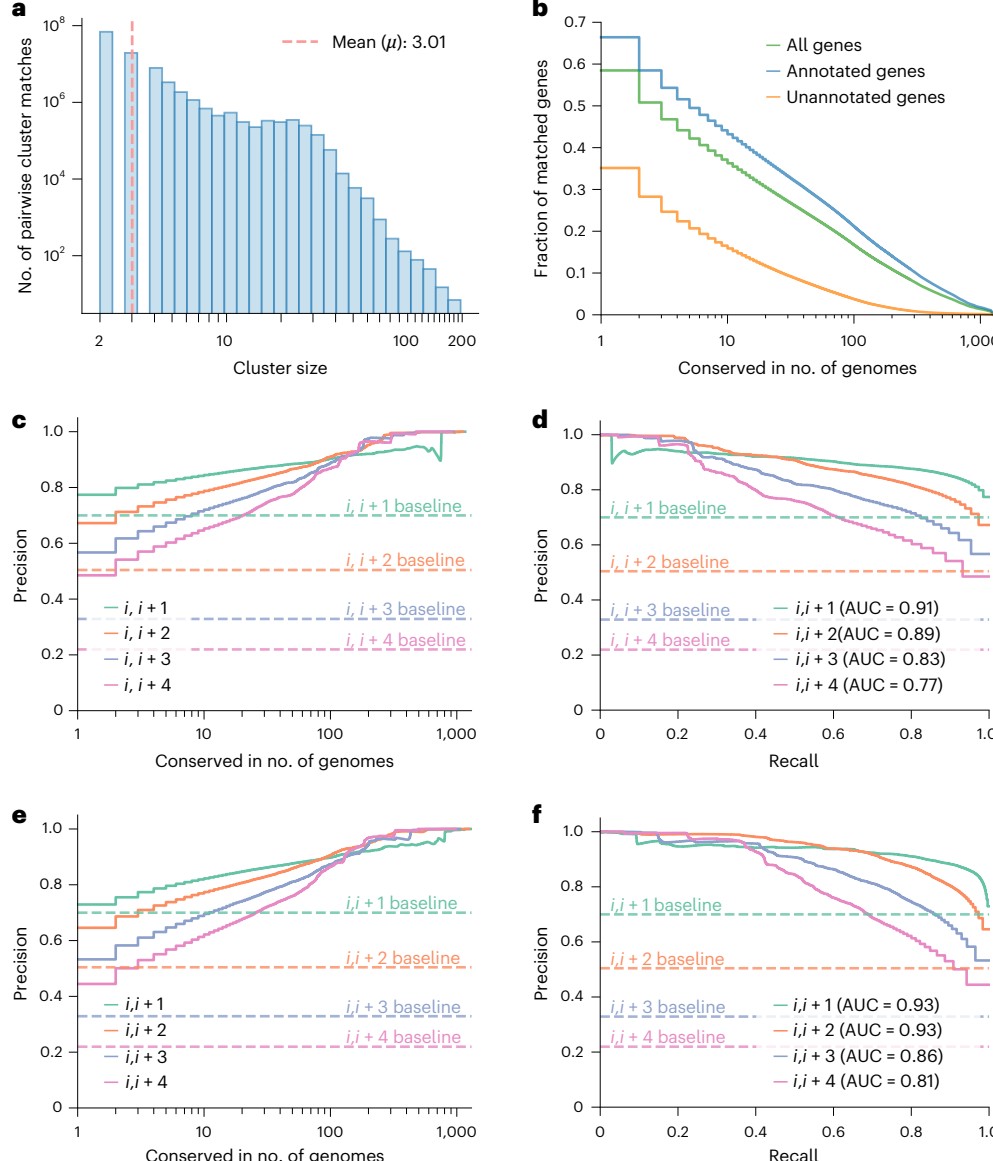

**Fig. 2 | Conservation of gene clusters identified by Spacedust predicts functional association. a**, Distribution of cluster sizes of all 106.6 million pairwise cluster matches among 1,308 bacterial genomes. **b**, Number of all (green), annotated (blue) and unannotated (orange) genes forming part of a cluster match in at least the number of genomes shown on the x axis. **c**,**e**, Precision of the functional association of gene pairs, separated by up to four genes in Spacedust cluster matches, versus the number of genomes in which the pair is conserved. True positive predictions are those gene pairs with the same KEGG module IDs. **c**, Foldseek+MMseqs search. **e**, Foldseek-only search with ProstT5. **d**,**f**, Precision versus recall of functional association of gene pairs separated by up to four genes. The analysis excludes ribosomal genes; see Extended Data Fig. 1 for analysis with ribosomal genes. **d**, Foldseek+MMseqs search. **f**, Foldseek-only search with ProstT5. AUC, area under the curve.

These genomes were selected such that they belong to different bacterial genera (Methods), to focus on detecting remote homology and globally conserved clusters across higher taxonomic ranks. This choice means species-specific and genus-specific gene clusters cannot be found in this analysis. We subjected all predicted genes (4.19 million) from the 1,308 genomes to an all-versus-all Foldseek+MMseqs2 search using Spacedust. The all-versus-all homology search and hit-filtering process took 72 h to complete on two servers with two 64-core AMD EPYC ROME 7,742 CPUs each, or 150 ms per genome–genome comparison (Supplementary Information), and the subsequent cluster detection required 51 min. The runtime of Spacedust scales quadratically with the total number of genomes and proteins in the genomes, owing to the all-versus-all search and cluster detection. The search yielded 321.2 million cluster hits in 106.6 million cluster matches, with an average of three genes per cluster match (Fig. 2a).

These pairwise cluster matches were subsequently grouped on the level of genome and genes, which yielded 72,483 nonredundant clusters comprising 2.45 million genes, representing 58% of the dataset. We classified 4.19 million genes based on their eggNOG-mapper annotations: 'annotated' if the gene was assigned a specific function (3.13 million, or 75% of the dataset), or 'unannotated' if the gene was labeled as 'hypothetical protein', 'protein with unknown function', or lacking any annotation (1.06 million, or 25% of the dataset). Notably, 66% of the annotated genes were found in nonredundant clusters present in more than one genome. Additionally, 35% of the unannotated genes were found in nonredundant clusters (Fig. 2b).

To evaluate the functional associations within the nonredundant clusters, we evaluated the congruence of KEGG module IDs for gapped gene pairs separated by up to four genes ($i, i + 1$)...($i, i + 4$; Fig. 2c and Extended Data Fig. 1). A gene pair is considered a true positive if

both genes share a common KEGG module ID, and false positive if not. The area under the precision–recall curve indicates that Spacedust identifies cluster matches with considerably higher accuracy than the baseline model, which assumes any neighboring gene pair $(i, i + x)$ to be functionally associated. Similarly, we assessed the functional association within the nonredundant clusters predicted by the Foldseek-only search mode (Fig. 2e,f), with the area under the precision–recall curve for gapped gene pairs $(i, i + 1)…(i, i + 4)$ slightly higher than that of the default Foldseek+MMseqs search mode.

### Global functional conservation of a cyanobacterium genome

To illustrate how Spacedust can support functional annotation of genomes, we took one example genome of a unicellular cyanobacterium *Synechocystis* sp. Pasteur Culture Collection (PCC) 6803 from the reference database with the 1,308 genomes. This genome comprises one chromosome (GenBank accession: BA000022.2) and four plasmids (AP004311.1, AP004312.1, AP004310.1, AP006585.1), totaling 3,551 protein-coding genes. All the detected clusters are visualized as an interactive cluster heat map (Extended Data Figs. 2–4). For better visibility, we zoomed in on a specific region spanning protein location indices 500 to 800 (Fig. 3a; corresponding to 0.0007% of the total dataset) and integrated functional annotation data obtained from eggNOG-mapper[51]. This allowed us to assign functions to many of the proteins. From this selected genomic region containing 300 genes, we identified three distinct cyanobacteria-specific clusters, indicative of functional conservation across related species. Additionally, we detected 21 clusters shared with other phyla (Supplementary Tables 1–3). Some clusters corresponded to single operons, while others spanned multiple operons.

Cluster 1 (Fig. 3b and Extended Data Fig. 5) comprises genes associated with photosystem II (PSII), the protein–pigment complex that drives oxygenic photosynthesis. The first two genes, rubredoxin and *ycf48*, are crucial for PSII activity and assembly. The remaining genes, *psbEFLJ*, form an operon encoding components of the core PSII complex. In many cases, *psbL* and *psbJ* are absent, possibly owing to poor conservation or the short length of their sequences.

Cluster 2 (Extended Data Fig. 6) forms an operon encompassing components of the phycobilisome complex rod, a large protein complex in cyanobacteria responsible for capturing sunlight and transferring energy to the photosynthetic reaction centers. The genes *cpcA* and *cpcB* encode two major subunits of the rod, while *cpcD*, *cpcC* and *cpcC2* encode linker components connecting the rod to the PBS core[52]. Conversely, homologous clusters in some other cyanobacteria only contain one copy of the *cpcC* gene, suggesting that *cpcC* and *cpcC2* might have been created by gene duplication. In some genomes, the genes are still colocalized despite the order of the genes being only partially conserved.

In cluster 3 (Extended Data Fig. 7), the first two genes are both annotated as *spkA* by eggNOG-mapper, encoding a eukaryotic-type serine/threonine protein kinase involved in signal transduction and mobility. Alignment with other clusters revealed gene fusion of these two genes in other cyanobacteria[53]. The third gene in the cluster is highly conserved in other genomes but could not be annotated using eggNOG-mapper.

### De novo identification of specialized gene clusters

To further assess the ability of Spacedust to identify conserved gene clusters, we focused on two categories of known, specialized gene clusters, antiviral defense systems and BGCs.

We used PADLOC (v1.1.0)[33] to identify all known antiviral defense systems in the 1,308 bacterial reference genomes. We removed any predicted region consisting only of single genes. Spacedust was able to recover 5,255 (95%) of 5,520 multi-gene defense system clusters detected by PADLOC (Fig. 4a,b), with 93% (4,888) of the clusters matching fully and 7% (367) matching partially to the PADLOC prediction.

Most partial cluster matches resulted from missing matches to one or two short genes at the edge of longer clusters such as CRISPR–Cas systems. For 73 of the 106 defense system types, more than 90% of all defense system clusters were discovered in their entirety by Spacedust (Fig. 4a), despite that restriction–modification type II clusters are the most abundant type of defense systems in the dataset yet most challenging for Spacedust to detect.

To evaluate Spacedust's ability to recover BGCs, we utilized a gold-standard dataset consisting of nine complete genomes available at the NCBI that were fully annotated with BGC and non-BGC regions[29]. We queried these nine genomes against our reference set of 1,308 bacterial genomes using Spacedust. We compared the results with three tools specialized in identifying BGCs, ClusterFinder[29], DeepBGC[30] (using a cutoff of 10% false positive rate as recommended) and GECCO[31]. As Spacedust returns all conserved clusters and not exclusively BGCs, it was not feasible to compare the precision of BGC detection based on all predictions. Therefore, we evaluated the F1 score, equal to the harmonic mean of precision and recall, for each of the annotated BGCs (Fig. 5). The precision is the fraction of genes in the overlapping predicted region that were annotated as BGC genes, and the recall is the fraction of genes in the annotated BGC region that were predicted as BGC region by the tool. Spacedust achieves higher F1 scores than ClusterFinder, DeepBGC and GECCO (Fig. 5a–c) owing to its higher precision than DeepBGC and GECCO and its higher recall than ClusterFinder (Extended Data Figs. 8 and 9). All tools failed to detect a few instances of BGCs that were identified by Spacedust. Figure 5d shows the cumulative distribution of the F1 score for the three tools. The average F1 score over all BGCs is 0.44 for ClusterFinder, 0.39 for DeepBGC, 0.43 for GECCO and 0.61 for Spacedust.

We used AntiSMASH (version 8)[28], a tool for profile-based BGC detection, to functionally annotate the genes as either 'biosynthetic-related' (biosynthetic, biosynthetic additional, transport, regulatory) or 'other genes'. We manually inspected the clusters reported by Spacedust, ClusterFinder, DeepBGC and GECCO. We observed that the regions reported by Spacedust often miss the transport and regulatory genes but cover the core and additional biosynthetic-related genes, sometimes with multiple short gene clusters within the annotated BGC region (Extended Data Fig. 10).

### Expansion of CRISPR–Cas subtype III-E single effector Cas7-11

Next, we investigated Spacedust's utility in identifying new instances of known gene cluster families. One such example is the recently discovered CRISPR subtype III-E, which comprises a single effector protein known as Cas7-11 (ref. 54). Notably, Cas7-11 is a protein fusing four Cas7 proteins with a putative Cas11-like protein. The fusion yields a single-protein programmable RNase that shows high sequence specificity and no evidence of collateral activity. Previous screening for Cas7-11 across bacterial genomic sequences led to the identification of subtype III-E systems in 17 loci.

To expand our knowledge of subtype III-E systems beyond the reported loci, we queried the proteins in the 17 loci reported by ref. 54 against the GTDB database[55]. Because we were unable to map a substantial portion of the query proteins and GTDB proteins to known structures, we used the MMseqs2 iterative search with three iterations to perform the homology search. We identified an additional seven instances of subtype III-E clusters in the GTDB database by demanding the presence of the gene encoding Cas7-11 (Fig. 6). In three out of seven genomes, all components of the respective system were identified, demonstrating the high sensitivity of the method.

### Spacedust Colab notebook

To facilitate the use of Spacedust for a broad user base, we have set up a Google Colaboratory environment, which allows users to easily run tests and reproduce results without requiring a local installation or configuration. We provide a comprehensive IPython notebook (ipynb file)

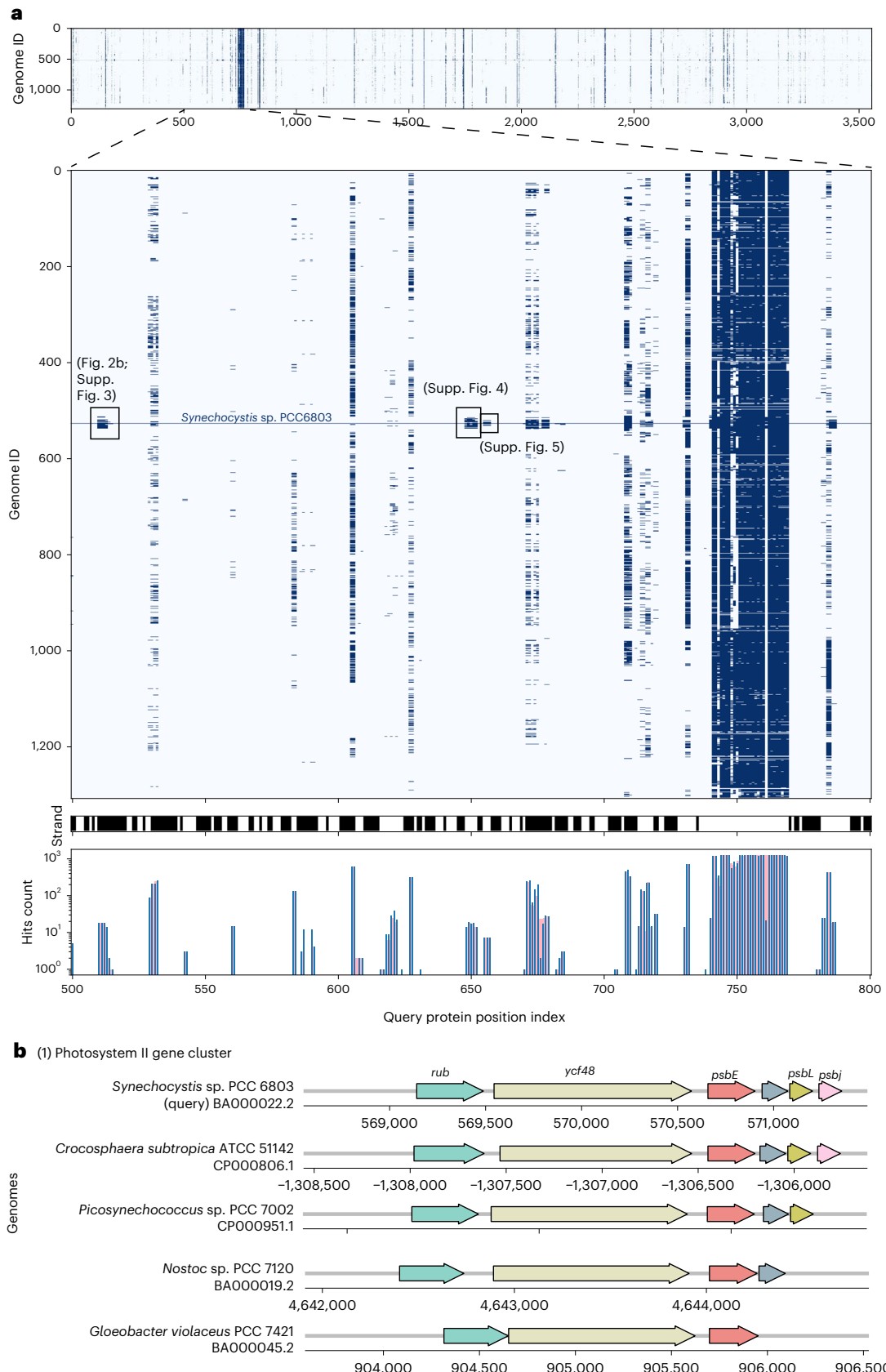

**Fig. 3 | Evolutionary conservation of gene clusters in an example cyanobacterium. a**, Zoomed view of clustered hits of *Synechocystis* sp. PCC 6803 (genome ID 527) against 1,308 bacterial reference genomes. Query proteins with location indices 500–800 on the genome shown. Top, Cluster heat map of the presence/absence of clustered hits across reference bacteria. Middle,

Transcription direction (black, forward; white, reverse). Bottom, Number of clustered hits per protein (blue) and hit pairs in the same gene cluster (pink). **b**, Example cyanobacteria-specific gene cluster 1. Gene names annotated by eggNOG-mapper are shown at the top. ATCC, American Type Culture Collection.

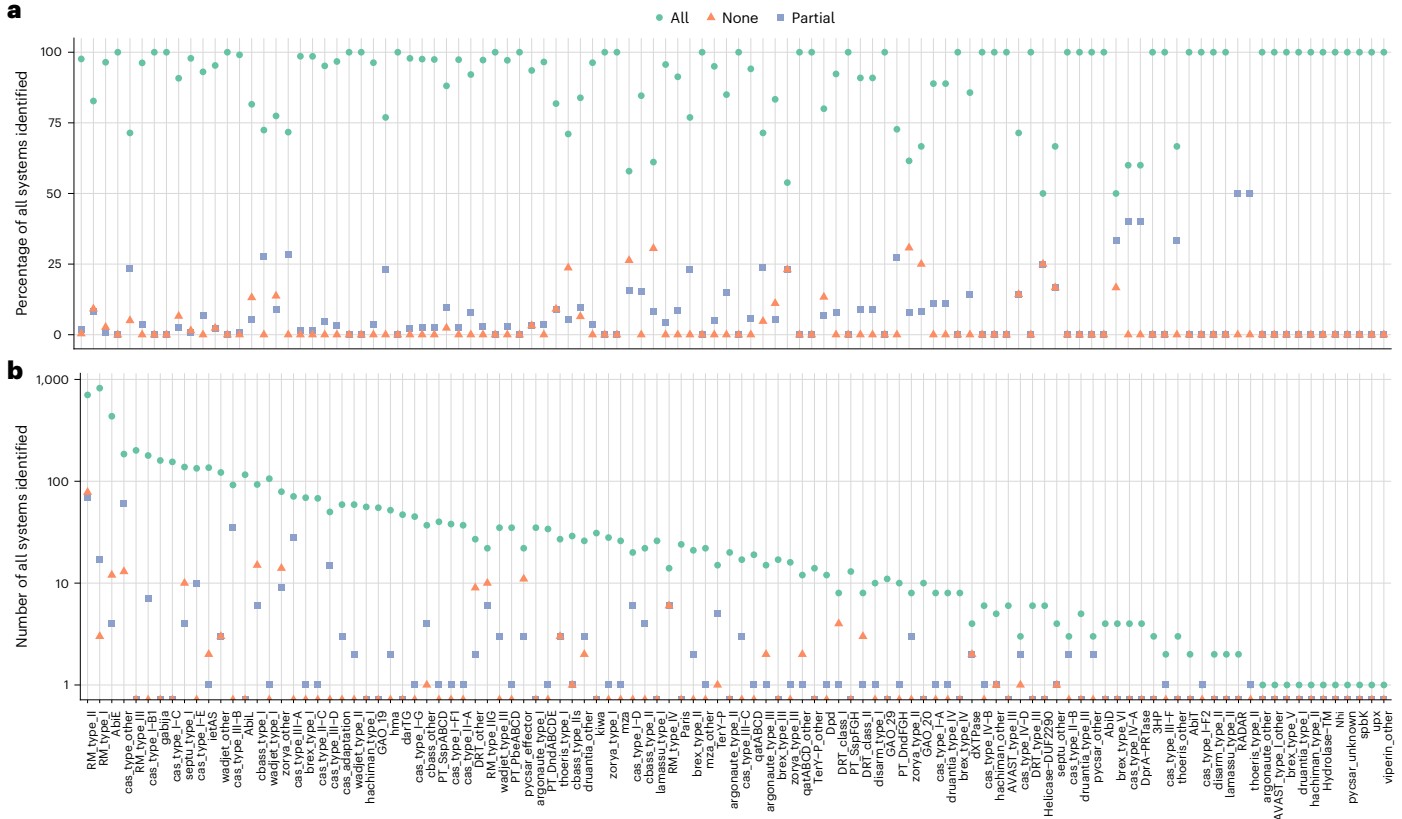

**Fig. 4 | Spacedust recovers the vast majority of antiviral defense systems predicted by specialized tools. a,b,** Percentage (**a**) and number (**b**) of multi-gene antiviral defense system clusters predicted by PADLOC that are also discovered by Spacedust in their entirety (green circle), partially (blue square) or missed (red triangle) within 1,308 bacterial genomes. 95% of all defense system clusters predicted by PADLOC were recovered, of which 93% are in full length.

that includes steps for installing all dependencies and databases, executing the program, and interactively visualizing the clusters. Within the Colab framework, users can either run Spacedust in an all-versus-all mode or annotate query genomes against a pre-compiled reference database with just a single click. With the help of interactive visualization, they can explore the evolutionary conservation of their genome of interest in other genomes at different resolutions and generate gene neighborhood plots for any gene clusters.

## Discussion

Exploiting the conservation of gene neighborhoods to predict functional association between genes is an old idea. So far, the main limitations have been (1) the rather low fraction of genes that are part of a conserved gene cluster[44], and (2) the low reliability of the inference of functional association.

Spacedust addresses limitation 1 in three ways. First, it finds homologous proteins using protein structure comparison with Foldseek, which is much more sensitive than the sequence search-based methods used so far. The increased sensitivity yields a higher number of conserved cluster matches between genomes (Extended Data Figs. 2–4). Second, owing to Foldseek's high speed and Spacedust's clustered search mode, it can analyze large sets of genomes in an all-versus-all fashion. The large number of genomes increases the chances that a gene will be part of a gene cluster that is conserved in another genome. Certainly, all gene clusters that are laterally transferred as a functional unit—such as BGCs[56]—will be detectable by Spacedust if a sufficiently large number of genomes is analyzed. Third, Spacedust does not require exact synteny but can find partially conserved neighborhoods (for example, Fig. 6). The success of these measures is demonstrated by the high fraction (58%) of genes that are part of a conserved gene

cluster among the 4.2 million proteins from the reference genomes of 1,308 bacterial genera (Fig. 2b), as well as the high sensitivities attained for the de novo discovery of antiviral defense systems and BGCs (Figs. 4 and 6).

Spacedust partially addresses limitation 2, the low reliability of functional association, by computing two novel P-value statistics for the significance of gene cluster conservation, one assessing the strength of positional clustering of the matched genes and the other assessing the degree of their strand and order conservation. These P values enable flexibility to find partially conserved gene clusters while still ensuring their statistical significance. Statistically significant conservation alone does not guarantee functional association, however. The fraction of gene pairs within a cluster match that are part of the same KEGG pathway can be as low as 50% when conserved between only two genomes (Fig. 3c), but it rises to over 80% for the 25% of the 4.2 million genes that are part of a cluster conserved in at least 50 of 1,308 genomes (Fig. 2b,c).

The relatively high fraction of KEGG-discordant gene pairs could be due to 'false false positives', which are functionally associated genes that are not labeled with the same KEGG pathway ID. However, we suspect that most discordant pairs are indeed not functionally associated. This observation was referred to as 'genomic hitchhiking' or 'carpooling' in ref. 46 and was later rationalized by Fang et al.[19]: Hitchhiking, the conservation of genomic neighborhoods containing groups of genes without obvious functional links between them, occurs mainly between core ('persistent') genes as a side effect of keeping functionally associated, accessory ('non-persistent') genes clustered together. As a consequence, accessory genes are less involved in hitchhiking. Hitchhiking generally limits the reliability of predicting functional association from conservation of gene neighborhoods.

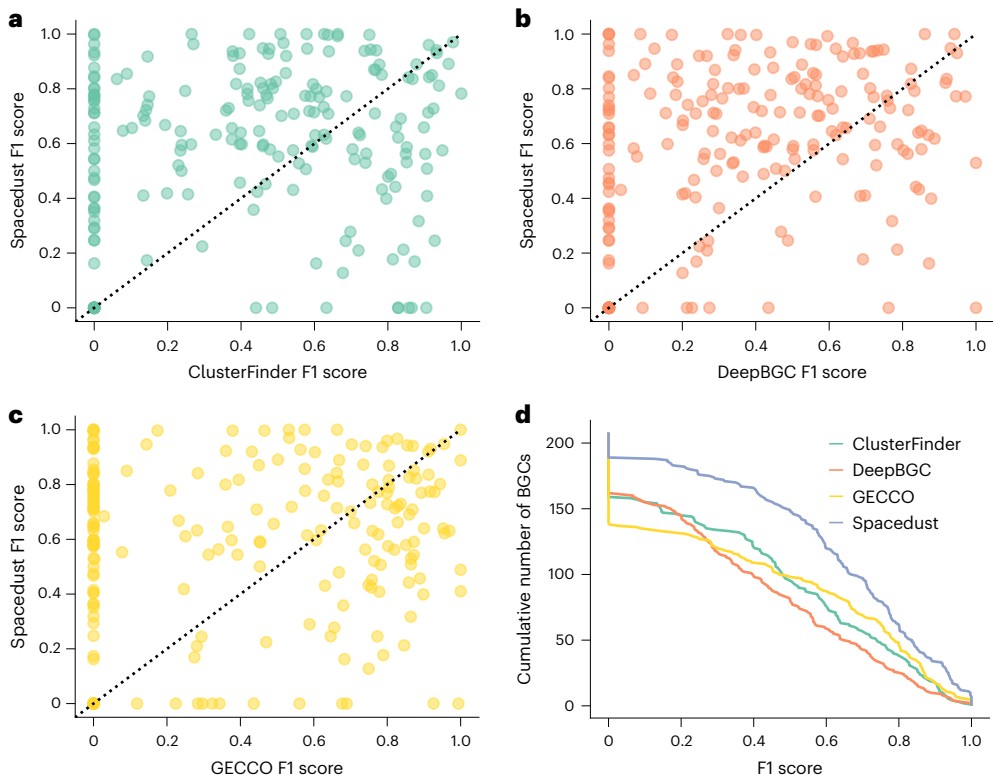

**Fig. 5 | Prediction of 207 manually annotated BGCs from nine genomes.**
**a**–**d**, For each of the 207 BGCs, we computed the F1 score as the harmonic mean of recall and precision. The recall for a BGC is the fraction of genes in the BGC that have been predicted by the tool, and the precision is the fraction of genes in the predicted region that overlap the annotated BGC. Scatterplots of F1 scores of ClusterFinder versus Spacedust (**a**), DeepBGC versus Spacedust (**b**) and GECCO versus Spacedust (**c**) for the 207 annotated BGCs. **d**, Cumulative distribution of the F1 scores for the 207 BGCs.

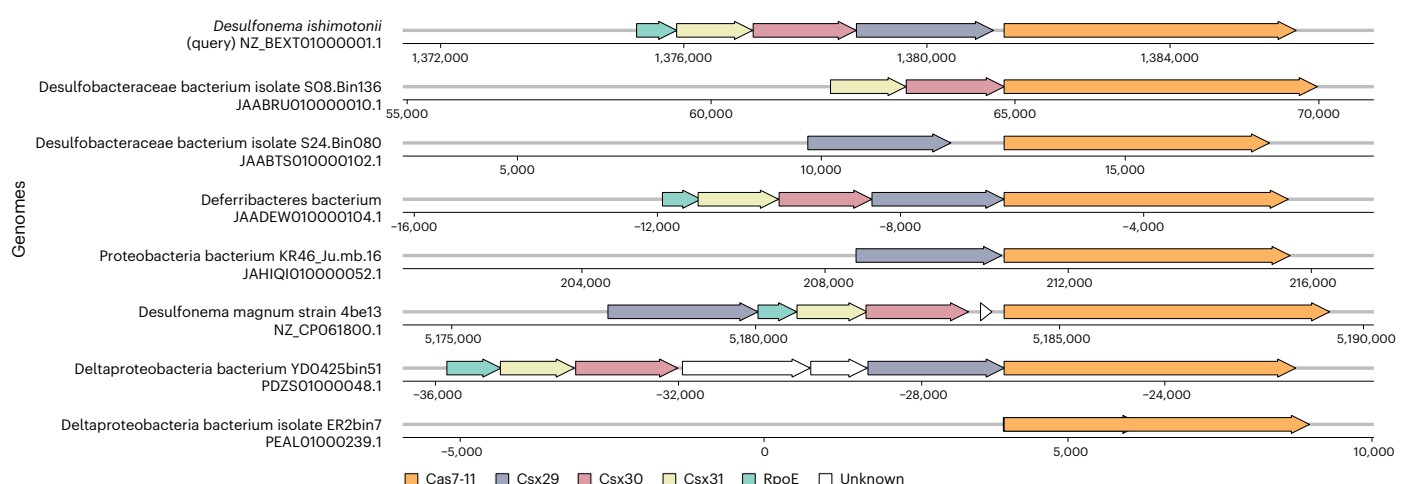

**Fig. 6 | Additional instances of CRISPR–Cas subtype III-E clusters identified in GTDB.** Visualization of the gene neighborhood of the clusters identified around the gene encoding Cas7-11 (orange arrow). *Desulfonema ishimotonii* (NZ_BEXT01000001.1), the representative of query III-E systems, is plotted as a reference to show the gene composition and order. Genes within the cluster boundary that could not be matched to Cas-related genes are colored in white.

To assess the sensitivity of Spacedust for finding functionally associated clusters of genes, we compared it with PADLOC, a tool specialized for finding antiviral defense systems. PADLOC relies on a hand-built library of approximately 3,800 HMMs to search for the proteins forming part of one of the 210 defense systems. Spacedust discovered de novo 95% of the defense system clusters annotated by PADLOC, of which 93% were discovered in their entirety (Fig. 4). Similarly, when assessing Spacedust on its ability to discover BGCs manually annotated in nine genomes, it performed better than GECCO, DeepBGC and ClusterFinder, which are trained on a large dataset of known BGCs (Fig. 5). In summary, Spacedust has similar sensitivity for de novo discovery of functional modules as dedicated tools trained for the discovery of a specific type of gene cluster. However, it is important to note that these specialized tools provide additional value by annotating clusters with specific biosynthetic classes or defense system types, which is not feasible with Spacedust.

The following four limitations of Spacedust need to be addressed in future work. First, partial conservation of a gene cluster in a certain

number of genomes predicts functional conservation with only moderate precision (Fig. 2c,d). We are working on an improved conservation score that takes the evolutionary divergence times between genomes into account. We also plan to increase precision by integrating operon predictions on all input genomes. Second, while the fast transformer tool ProstT5 can reliably predict three-dimensional interaction (3Di) sequences for well-studied reference bacterial proteins, its accuracy is less consistent for viral and metagenomic sequences. Although we provide the ProstT5 model to enable full Foldseek structure searches as an alternative to mapping precomputed structures, we emphasize this limitation to guide users in selecting appropriate modes based on their dataset. We are actively working to update the model and will ensure users can easily download the improved version once available. Third, Spacedust cannot find protein members of functional modules encoded outside a conserved gene cluster[26]. We will address this limitation by applying Spacedust for building a database of module-specific protein families and profile HMMs with HMM-specific acceptance thresholds (similar to Pfam[57]), which should allow us to identify also positionally isolated members of functional modules with high specificity. Another limitation of Spacedust is its quadratic scaling of runtime with the total number of genomes and proteins in the genomes to be analyzed, caused by the quadratic time complexity of the all-versus-all comparison of proteomes and of the cluster detection algorithm.

Positional orthologs, that is, orthologs that also have conserved gene neighborhoods, are under much stronger evolutionary constraints than orthologs without gene neighborhood conservation[58]. Therefore, we expect functional module-specific protein families to show high functional conservation and to become highly useful for automatic functional annotation using a comprehensive profile HMM database of such modules. We also plan to use Spacedust for the systematic discovery of uber-operons or extended gene neighborhoods[46,59,60], sets of genes that tend to co-occur in each other's neighborhood more often than by chance and that tend to participate in the same or related processes in the cell.

In conclusion, Spacedust is a sensitive and fast tool for finding conserved gene clusters in large numbers of genomes. It can be used for the large-scale discovery of modules of functionally associated genes in prokaryotic and viral genomes and metagenome-assembled genomes, as demonstrated here with various examples. Its de novo approach and high sensitivity make it particularly interesting for the discovery of novel types of functional modules. It can visualize conserved clusters in a query genome across hundreds of target genomes (Fig. 3a). Its tabular output facilitates its integration into current genome annotation pipelines. It can thereby accelerate the identification of the functional capabilities of the millions of prokaryotes and viruses that live in and on our bodies and populate all natural environments.

## Online content

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

## Methods

### Spacedust workflow

**Input.** Spacedust accepts genomic sequences as multiple FASTA files, each containing a single prokaryotic genome or metagenome-assembled genome. Users can either predict the protein-coding sequences from the input genome using Prodigal (v2.6.3)[61] or provide the corresponding GFF3 annotation files of protein-coding regions. Contigs belonging to the same assembly should be contained in a single FASTA file. All protein-coding regions are extracted and translated. For each protein sequence, the location index, strand and nucleotide coordinates are stored. For all-versus-all comparisons, only one set of genomes is required. For query-to-reference comparisons, a custom database can be built analogously, or a pre-compiled reference database can be automatically downloaded by Spacedust.

**Mapping to structure database.** To enable structure comparisons, query protein sequences are mapped to the reference structure database provided by Foldseek[48]. Currently, Foldseek supports several structure databases such as AlphaFold (UniProt, Proteome, Swiss-Prot), Protein Data Bank and ESMAtlas30. For each protein, in addition to the amino acid sequence, the structure information, including the 3Di sequence and the $C_\alpha$ coordinates, are stored in the Foldseek database. Each query protein sequence is searched against the amino acid sequences of the structure database using MMseqs2 (ref. [50]) with a stringent sequence identity cutoff of 0.9 and sequence coverage cutoff of 0.9 (`--min-seq-id 0.9 -c 0.9`), and the respective 3Di sequence and the $C_\alpha$ coordinates of the best match are retained to build a Foldseek-compatible database. Alternatively, Foldseek supports translation between protein sequences and 3Di sequences using the ProstT5 protein language model[62], with which a Foldseek-compatible database can be created for all query protein sequences.

**Homology search.** Spacedust conducts a sensitive search of all query proteins against target proteins using Foldseek and/or MMseqs2. If a Foldseek-compatible database is available, the mapped or translated structure sequences will be used in a Foldseek search, while any remaining unmapped sequences are searched using MMseqs2. The $E$-value cutoff is set at 0.001, and the query sequence coverage is set at 0.8 (`-e 0.001 -c 0.8 --cov-mode 2`). The results from both searches are merged. Users can also opt for an iterative profile (PSI-BLAST like) search by specifying the number of iterations.

**Clustered search.** Searching against a large sequence or structure database is a time-consuming and memory-demanding task. To improve the search speed while maintaining high sensitivity, we implemented a clustered search workflow similar to the strategy used in ColabFold[63]. The query sequences are searched against the consensus sequence or structure of the clustered version of the reference database. For each query hit to a consensus sequence, we realign the query to its respective cluster members and expand the search results. An additional advantage is the higher sensitivity attained using cluster consensus sequences. The clustered search approach results in a fourfold speed-up, because only 1 million cluster consensus structures are searched instead of 3.8 million structures from the bacterial reference database.

**Hit filtering and grouping.** Each protein in each of the genomes in the query set is searched through the proteins in each of the target genomes $T$ with $N_T$ proteins. For each query protein, the hit in $t$ with the lowest $P$ value $p$ is identified. If $p \leq p_0$ (default value $10^{-7}$), we compute the probability that in a comparison with $N_T$ nonhomologous proteins no $P$ value will be below $p$, given by the first-order $P$-value statistics, $p_{bh} = 1 - (1-p)^{N_T}$. The best hits between the pairs of query and target genomes are then grouped for subsequent cluster detection.

**Cluster detection.** For each pair of genomes, Spacedust uses a probabilistic approach to identify conserved neighborhoods of genes, which are clusters of homologous hits with partially conserved clustering and ordering. We assess the conservation of gene neighborhood by combining two $P$-value statistics, a clustering $P$ value and an ordering $P$ value. Given any cluster of hits $\mathcal{C}$, we can compute the number of hits $k$. The span $m$ is the maximum of the number of genes (including unmatched ones) in the query genome cluster and in the target genome cluster. We also define $q_0$ (default value $10^{-3}$) as the probability for an arbitrary protein $q$ to hit an arbitrary protein $t$ in $T$. The clustering $P$ value of the given cluster is the probability to observe a cluster of size $k$ each with a probability of $q_0$ within a square of span $m$ (Supplementary Information), as given by equation (1):

$$p_{\text{clu}}(\mathcal{C}) \approx \frac{m!^2}{(m-k)!^2 k!} q_0^k. \tag{1}$$

The ordering $P$ value assesses the statistical significance of the conservation of order and strandedness between two cluster matches. We define the directionality and ordering statistic $n \in \{0, 1, 2, \dots\}$ as the number of neighboring query protein pairs that are also direct neighbors and whose relative orientation is conserved. The ordering $P$ value is the probability to observe in a randomly occurring cluster match with $k$ matched proteins at least $n$ neighboring pairs with conserved order and strandedness. In the methods section of the Supplementary Information, and as shown in equation (2), this $P$ value is:

$$p_{\text{ord}}(\mathcal{C}) = \frac{1 - n/k}{2^n n!}. \tag{2}$$

Both statistics are independent of each other and of the strength of individual pairwise sequence homology and thus should improve the specificity of the search. Additionally, both statistics are not influenced by the size of the query and target sets, making them suitable for fragmented contigs with small numbers of genes.

The clusters are detected with a greedy agglomerative hierarchical clustering algorithm. Because the two $P$ values are independent random variables under the null model, we can combine them using the product of $P$ values $p := p_{\text{clu}}(\mathcal{C}) p_{\text{ord}}(\mathcal{C})$[64], which yields the cluster match $P$ value, $p \times (1 - \log p)$. We define a cluster score $S(\mathcal{C})$ as negative logarithm of the cluster match $P$ value: $S(\mathcal{C}) = -\log p_{\text{clu}}(\mathcal{C}) - \log p_{\text{ord}}(\mathcal{C}) + \log (1 - \log p_{\text{clu}}(\mathcal{C}) - \log p_{\text{ord}}(\mathcal{C}))$. The greedy agglomerative hierarchical clustering algorithm first treats each hit as a singleton cluster, and iteratively merges hits with the highest cluster match score satisfying the clustering criteria. Users can adjust the stringency by defining different clustering criteria, such as the maximum number of gaps (non-cluster genes) allowed and the minimum number of genes in a cluster. The probabilistic nature of the algorithm accounts for micro-rearrangements between genomes, gene insertions/losses and misannotated genes.

**Output.** Spacedust outputs a tab-separated text file. Each reported cluster consists of one summary line followed by multiple lines, one line for each pairwise hit. The summary line starts with '#': a unique cluster identifier, query genome accession, target genome accession, cluster match $P$ value (joint $P$ value of clustering and ordering), multi-hit $P$ value and number of hits in the cluster. Each following line describes an individual member hit of the cluster in MMseqs2 alignment-result-like format with the following columns: query protein accession, target protein accession, best-hit $P$ value $p_{bh}$, sequence identity, pairwise $E$-value, query protein start, end and length, target protein start, end and length, and alignment traceback string.

### Bacterial reference database

The bacterial reference database was assembled from the KEGG GENOME collection[65], which comprises 7,167 complete bacterial

genomes. The genomes were downloaded from NCBI GenBank in September 2022. Genomes were filtered for redundancy using pairwise average amino acid identities (AAIs). Specifically, we used Mash (v2.3)[66] to perform all-versus-all alignments using the amino acid alphabet (`-a`) with default parameters and computed the pairwise AAI as (1 − Mash distance). Next, genomes with AAIs of at least 70% were clustered using SciPy's hierarchical clustering function[67,68], and the longest sequence within each cluster was selected as the representative. The threshold roughly corresponds to genus-level clustering, meaning the representatives belong to different bacterial genera[69]. This resulted in 1,308 representative genomes that make up the reference database. We predicted the protein sequences using Prodigal v2.6.3 and constructed the Foldseek-compatible database as described above. Around 90.9% (3.8 million of 4.2 million) of the protein sequences could be mapped to a structure in the AlphaFold database. Comprehensive functional annotation of all protein sequences is included using eggNOG-mapper (v2.0)[51] with MMseqs2 search and default parameters.

### Reducing the size of the reference database

For large reference databases, the search step's memory requirements would make local runs infeasible. The reference genomes, even after the AAI-based redundancy filtering step, still contain sequence redundancy. Therefore, we further reduced the size of the database by clustering the protein and structure sequences. We clustered the protein sequences with MMseqs2 at 70% sequence identity and 80% bidirectional coverage (`--min-seq-id 0.70 -c 0.8 --cov-mode 0`). For structures, we first clustered the protein sequences at 30% sequence identity and 90% bidirectional coverage with MMseqs2 (`--min-seq-id 0.30 -c 0.8 --cov-mode 0`), and then further clustered the structure sequences with Foldseek without the sequence identity threshold but with 90% bidirectional coverage and an *E*-value of less than 0.01. Spacedust provides a search mode `--profile-cluster-search`. Under this search mode, Spacedust only performs MMseqs2 and Foldseek searches against the cluster consensus sequences and then expands to other members of the cluster to not lose hits.

### Reporting summary

Further information on research design is available in the Nature Portfolio Reporting Summary linked to this article.

### Data availability

Data used in this work were obtained from public sources and are freely accessible. The bacterial genome dataset was assembled from the KEGG GENOME collection (https://www.genome.jp/kegg/tables/br08606.html) downloaded from NCBI GenBank in September 2022. The protein structure database used for mapping was compiled from the AlphaFold database (https://alphafold.ebi.ac.uk/). The genomes and datasets used for the analysis in Fig. 5 and Extended Data Figs. 8–10 are publicly accessible via the supplementary material of Hannigan, G. D. et al.[30]. The genome used as a query in Fig. 6 is available from NCBI GenBank under accession number NZ_BEXT01000001.1. The target database is publicly available from GTDB (https://gtdb.ecogenomic.org/).

### Code availability

Spacedust is implemented in C++ and is available as an open-source (GPLv3), user-friendly, command-line software for Linux and macOS. The Spacedust source code, compilation instructions and a user guide are available at https://github.com/soedinglab/Spacedust/. The dataset of the 1,307 representative bacterial genomes and scripts to reproduce the search and visualize results are available at https://wwwuser.gwdg.de/~compbiol/spacedust/.

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

### Acknowledgements

We thank M. Steinegger for the suggestion to use Foldseek in Spacedust. We thank H. Su and E. L. Karin for their valuable feedback on the implementation and insightful comments on the manuscript. We used the Scientific Compute Cluster at GWDG, the joint data center of the Max Planck Society (MPG) and University of Göttingen. R.Z. acknowledges support by the IMPRS Genome Science graduate school. The work was supported by the BMBF CompLifeSci project horizontal4meta. M.M. acknowledges support from the National Research Foundation of Korea (NRF) (grant RS-2023-00250470). R.Z. was supported by the International Max Planck Research School for Genome Science.

### Author contributions

R.Z. and J.S. designed the Spacedust algorithm, benchmarks and biological applications. R.Z. and M.M. developed the software. R.Z. performed benchmarks and generated figures. R.Z. and J.S. wrote the manuscript. All authors read and approved the final manuscript.

### Funding

### Competing interests

The authors declare no competing interests.

### Additional information

**Extended data** is available for this paper at https://doi.org/10.1038/s41592-025-02816-x.

**Correspondence and requests for materials** should be addressed to Johannes Söding.

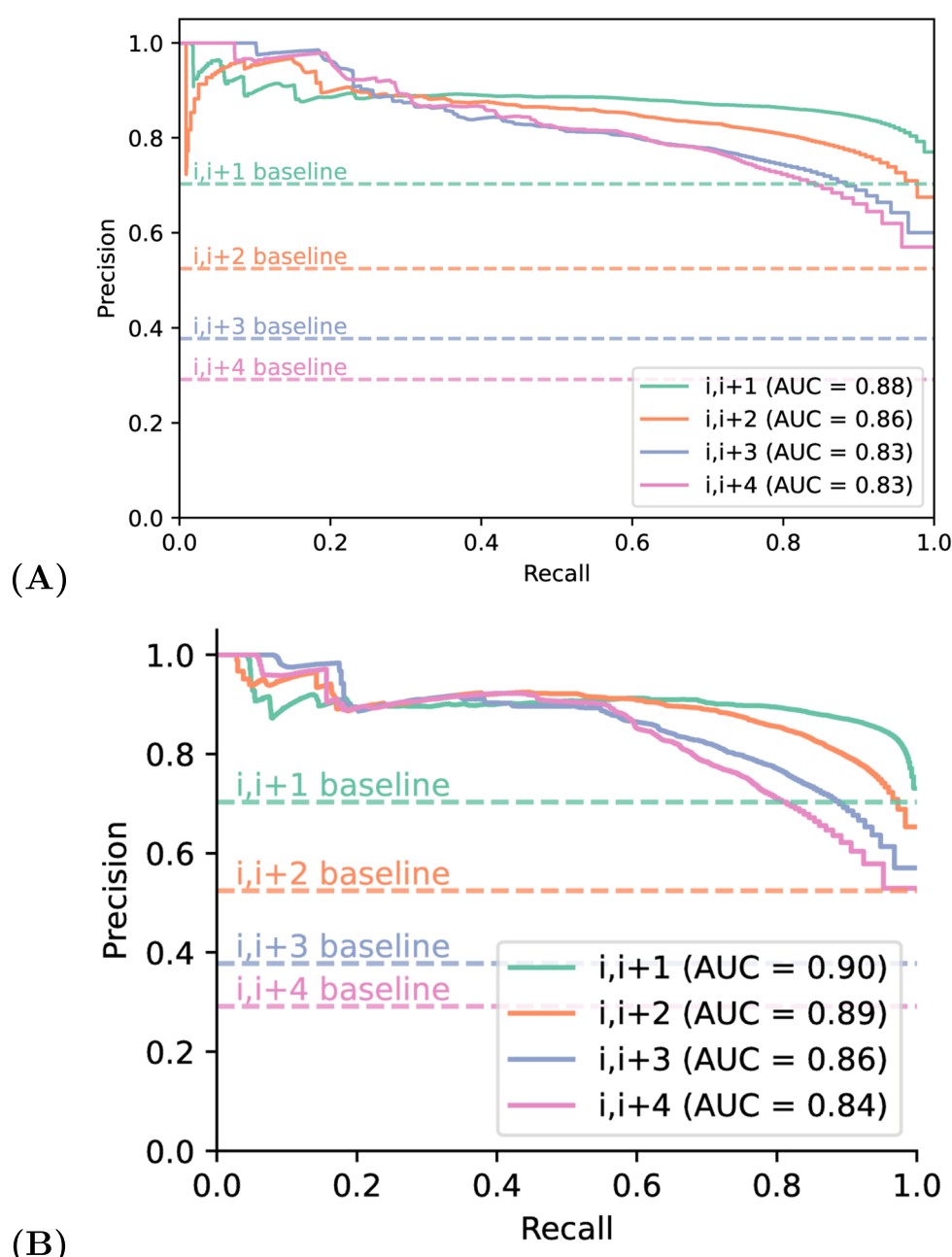

**Extended Data Fig. 1 | Precision-recall (PR) of functional association of non-redundant conserved clusters.** (including the ribosomal genes) for (**A**) Foldseek+MMseqs search and (**B**) Foldseek- only search with 3Di sequences predicted by ProstT5, assessed by congruence of KEGG module IDs of Spacedust cluster matches for all gene pairs separated by up to 4 genes (i,i+1),…, (i,i+4).

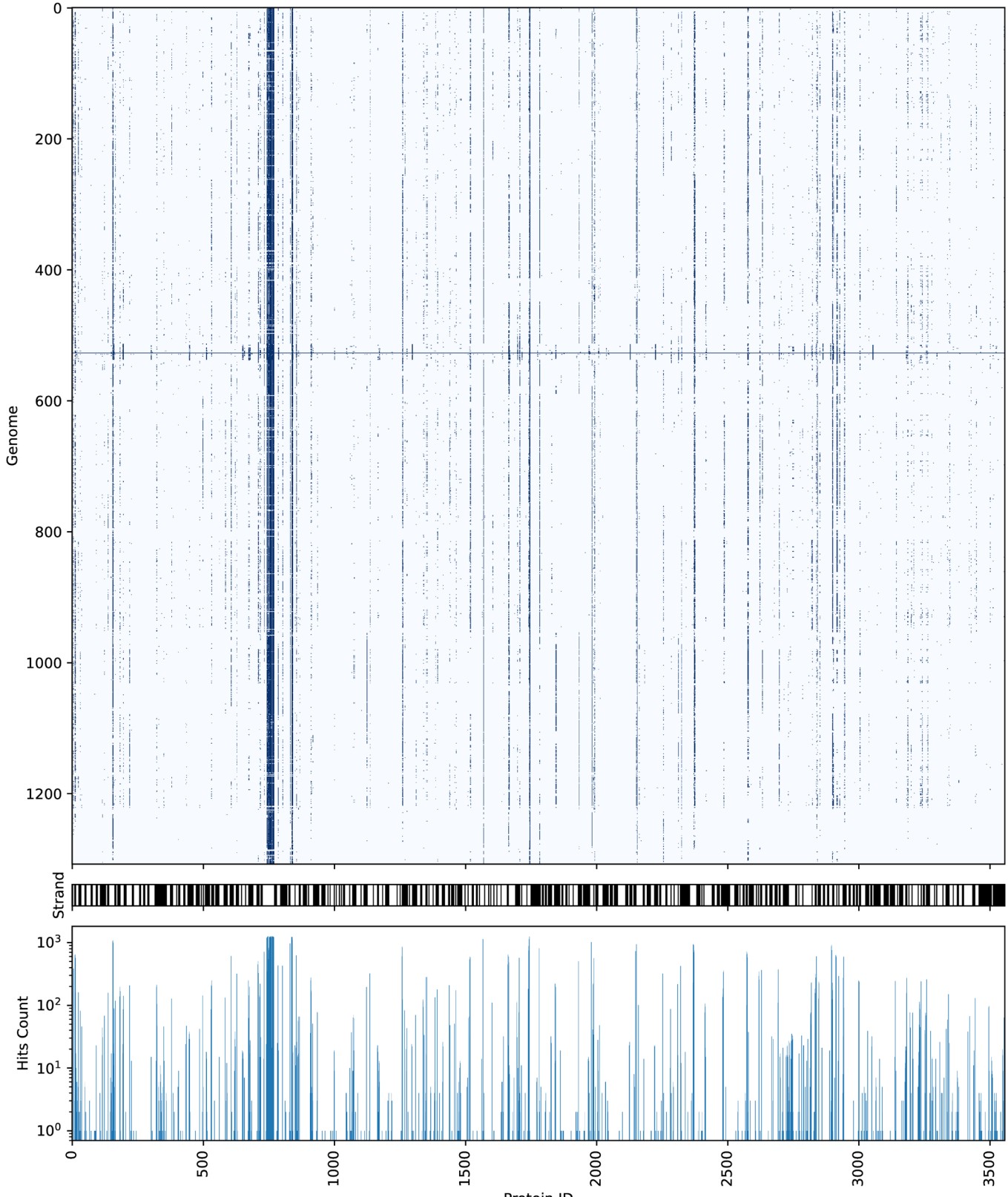

**Extended Data Fig. 2 | Evolutionary conservation of gene clusters in a cyanobacterium Synechocystis sp. PCC6803.** Clustered hits of Synechocystis sp. PCC6803 (Genome ID 527) against 1308 bacterial reference genomes using Spacedust Foldseek+MMseqs2 search.

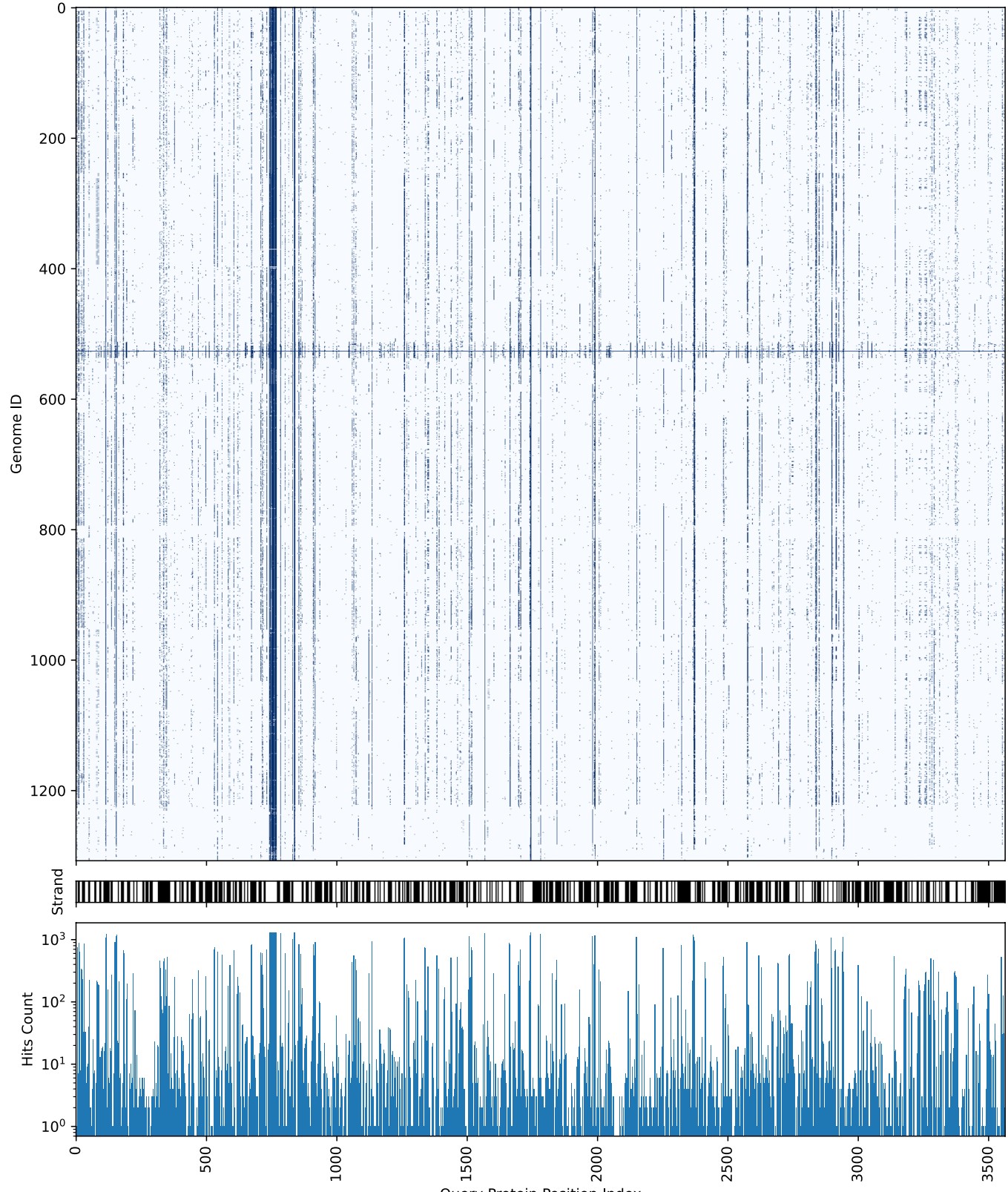

**Extended Data Fig. 3 | Evolutionary conservation of gene clusters in a cyanobacterium Synechocystis sp. PCC6803 (ProstT5).** Clustered hits of Synechocystis sp. PCC6803 (Genome ID 527) against 1308 bacterial reference genomes using Spacedust Foldseek-only search with 3Di sequences pre- dicted by ProstT5.

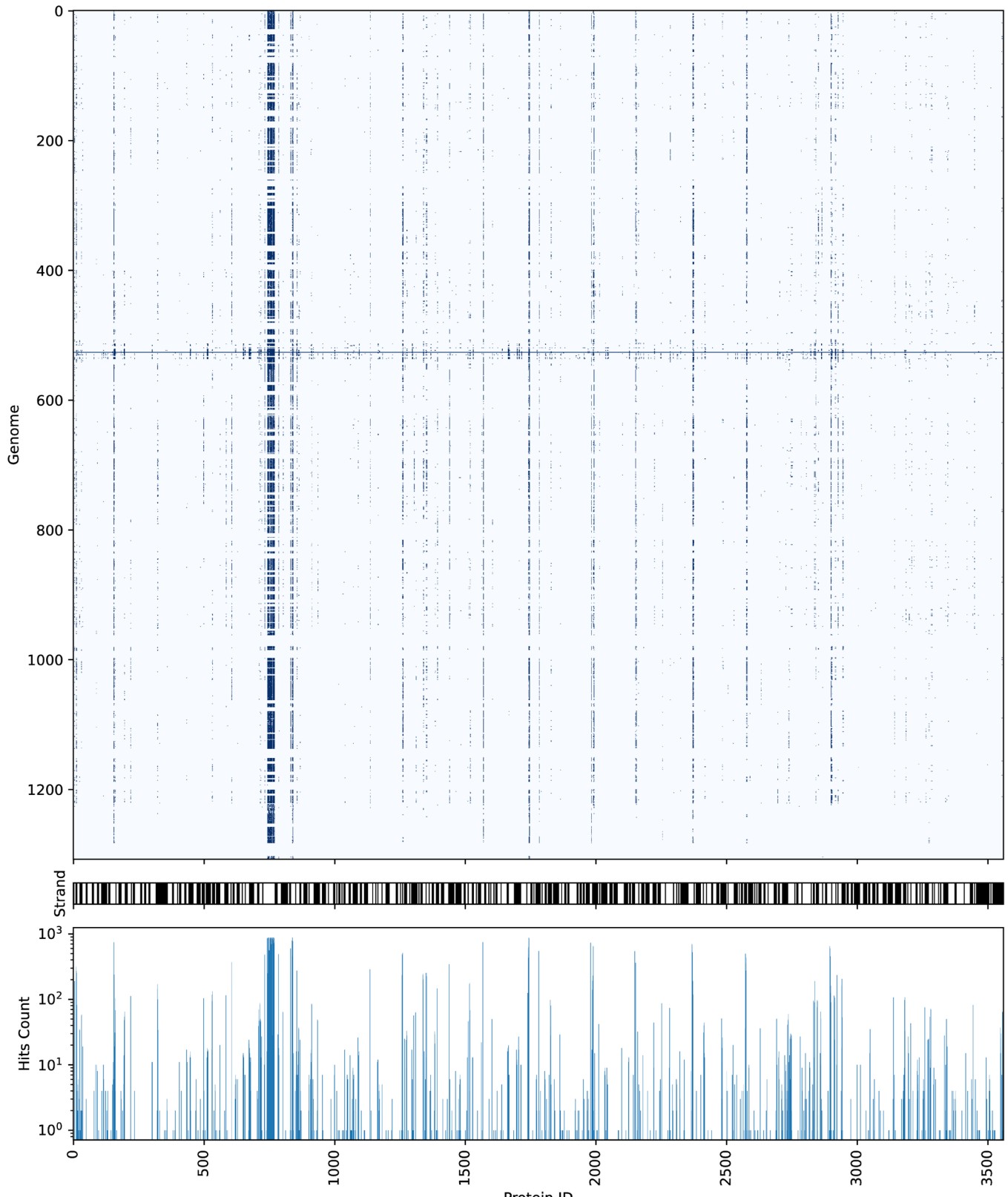

**Extended Data Fig. 4 | Evolutionary conservation of gene clusters in a cyanobacterium Synechocystis sp. PCC6803 (MMseqs2).** Clustered hits of Synechocystis sp. PCC6803 (Genome ID 527) against 1308 bacterial reference genomes using Spacedust MMseqs2 search.

## (1) Photosystem II gene cluster

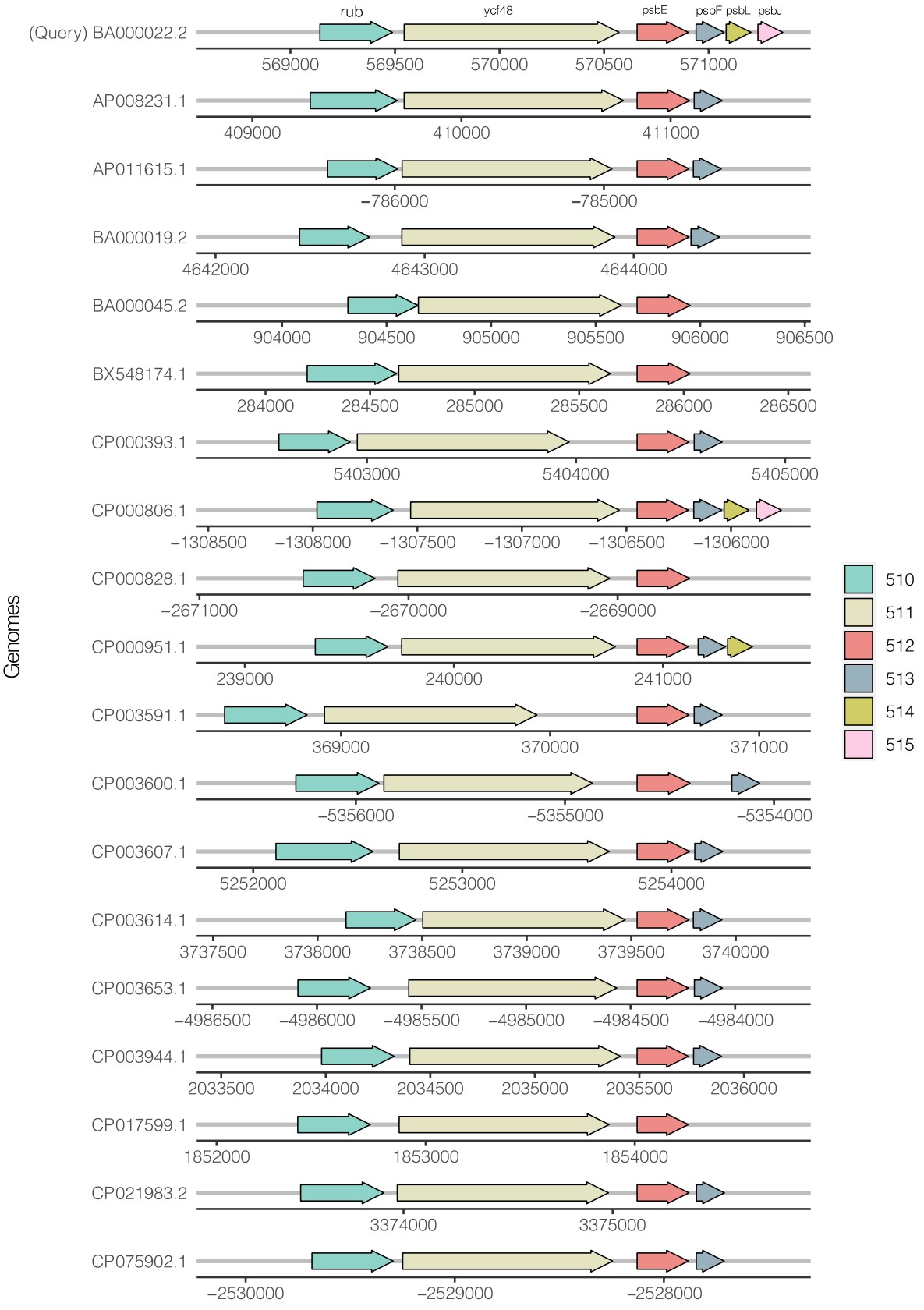

**Extended Data Fig. 5 | Gene neighborhood of Cyanobacteria-specific cluster 1.** (Protein ID 510- 515), centered around protein 512.

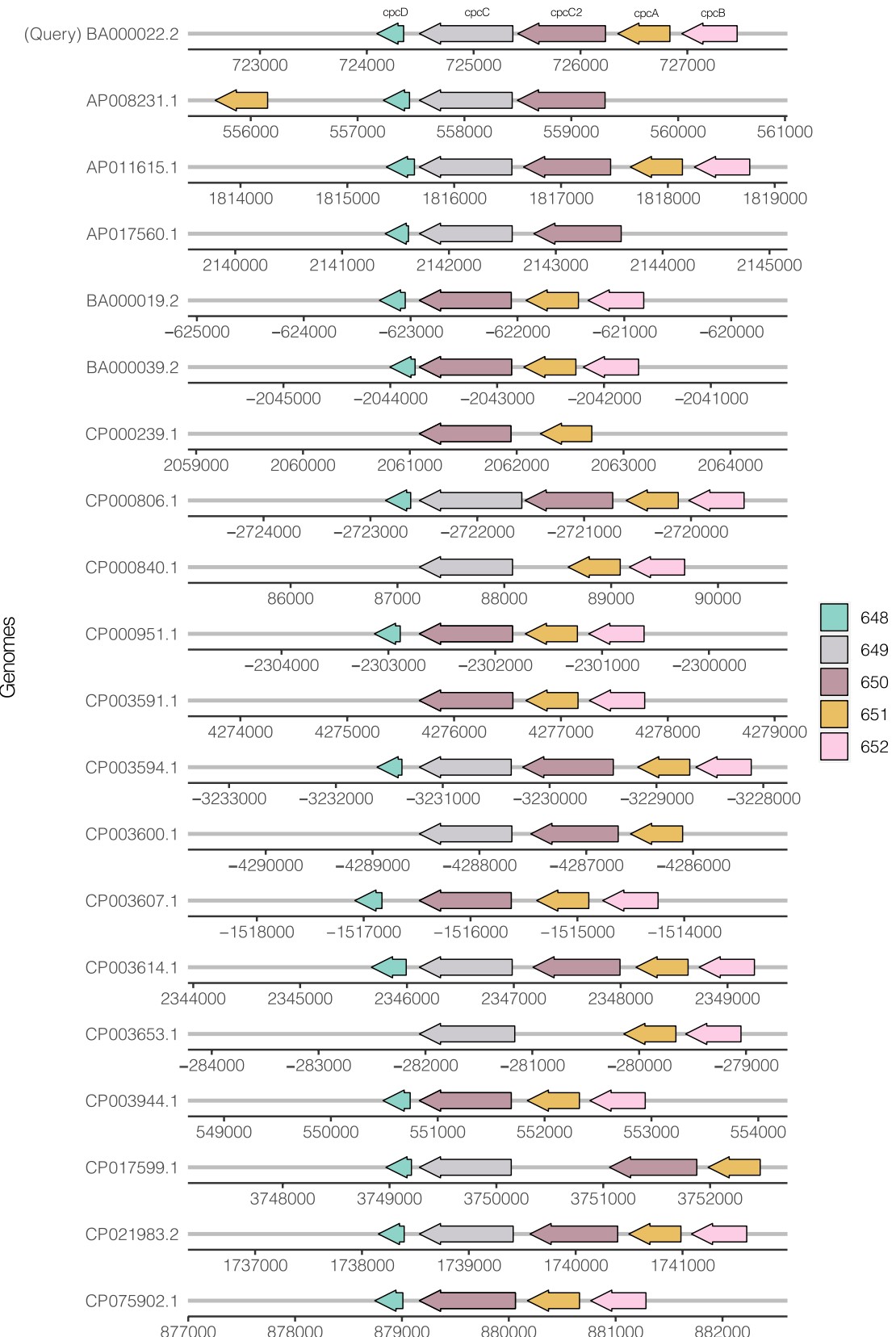

**Extended Data Fig. 6 | Gene neighborhood of Cyanobacteria-specific cluster 2.** (Protein ID 648- 652), centered around protein 649.

## (3) Eukaryotic-type Ser/Thr protein kinase & unknown protein

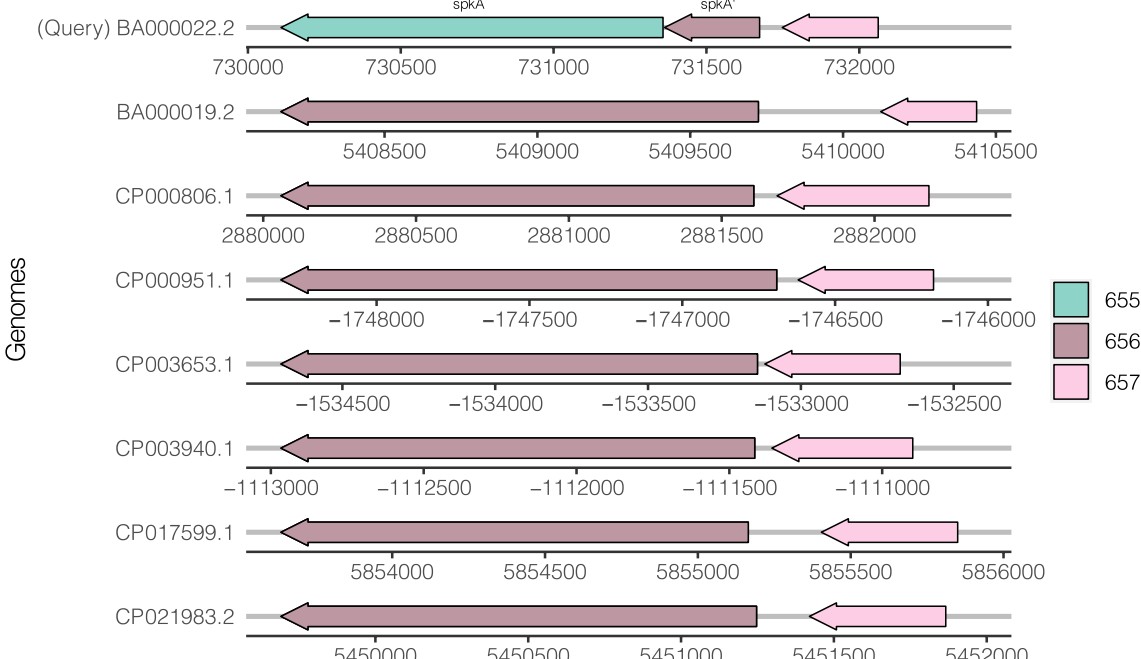

**Extended Data Fig. 7 | Gene neighborhood of Cyanobacteria-specific cluster 3.** (Protein ID 655- 657), centered around protein 655.

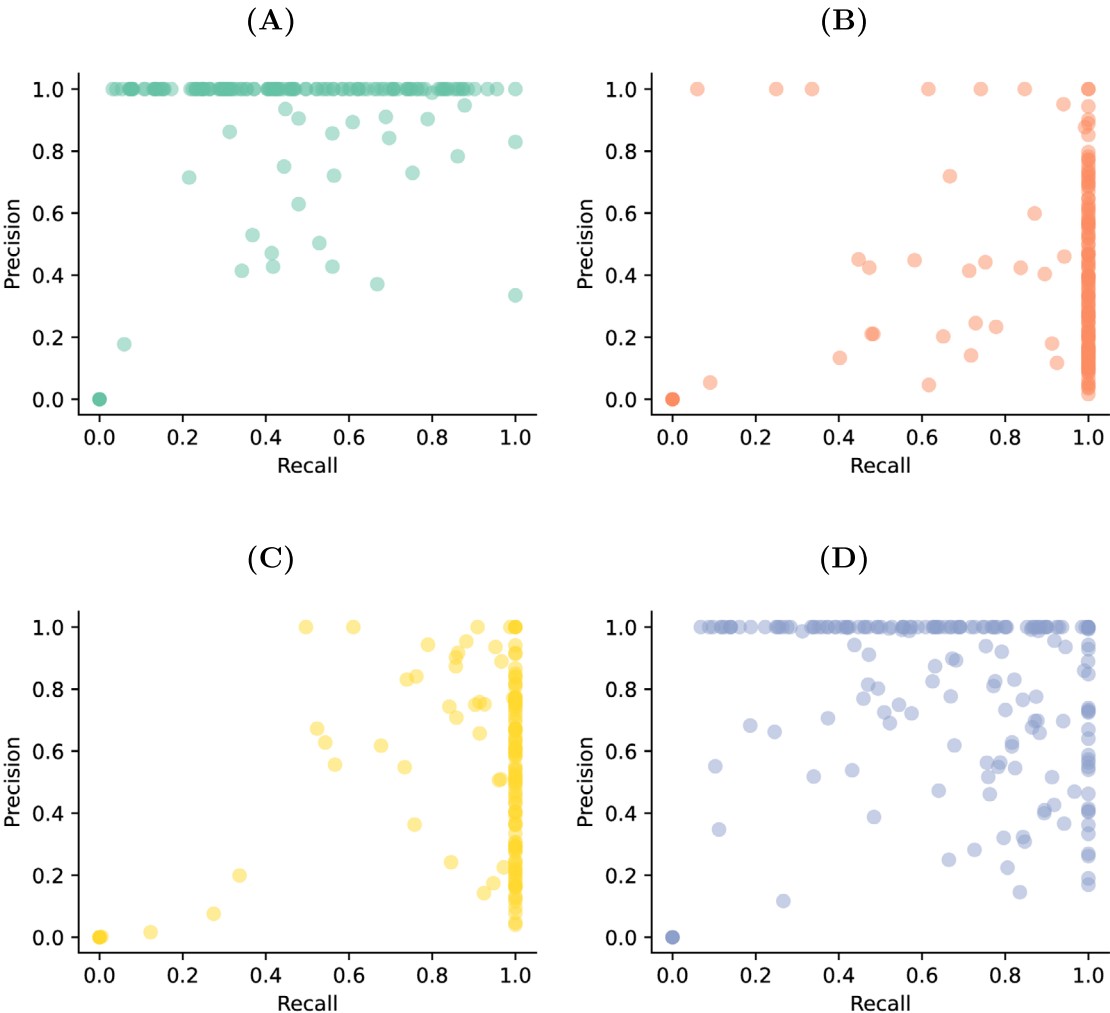

**Extended Data Fig. 8 | Scatter plots of precision versus recall for the 207 annotated BGCs.** for (**A**) Clusterfinder, (**B**) DeepBGC, (**C**) GECCO and (**D**) Spacedust.

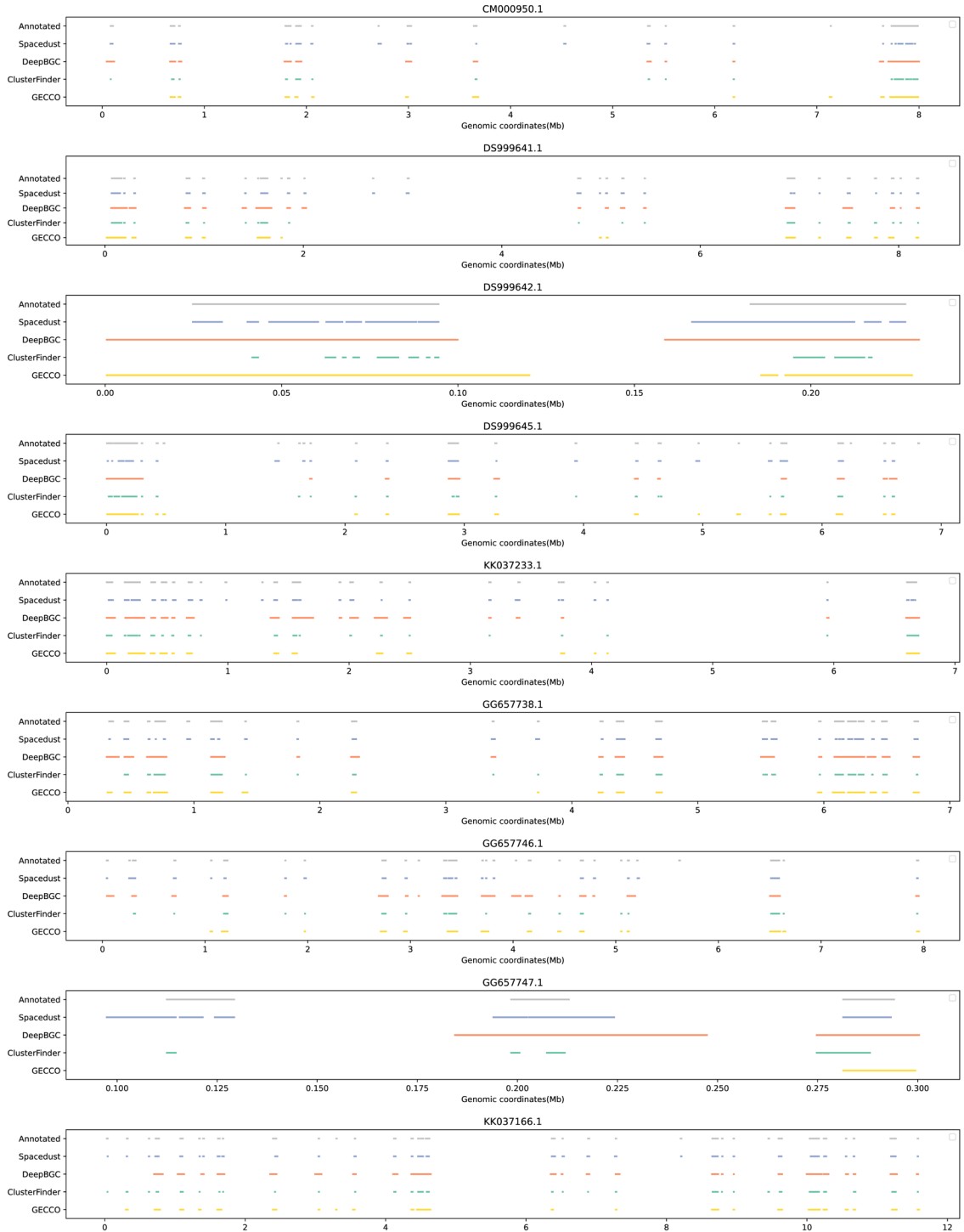

**Extended Data Fig. 9 | Contig view of 9 reference genomes with genomic regions.** predicted by ClusterFinder (green), DeepBGC (orange), GECCO (yellow) and Spacedust (blue) overlapping with the annotated BGCs (grey).

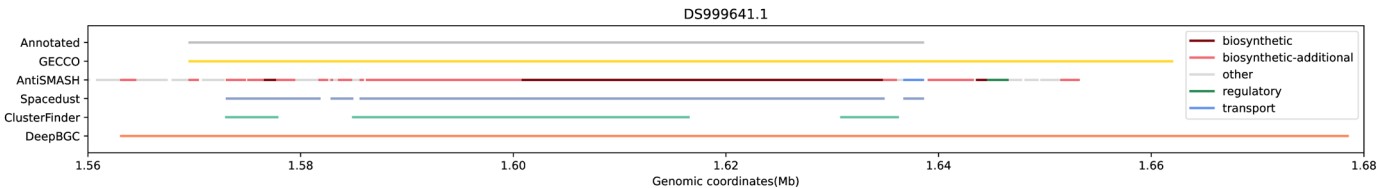

**Extended Data Fig. 10 | Example BGC regions (DS999641.1).** identified by ClusterFinder (green), DeepBGC (orange), GECCO (Yellow) and Spacedust (blue), superimposed upon annotated BGCs (grey) along with AntiSMASH (version 8) predictions and functional categories.

# Reporting Summary

## Statistics

For all statistical analyses, confirm that the following items are present in the figure legend, table legend, main text, or Methods section.

| n/a | Confirmed | |
|---|---|---|
| ☐ | ☒ | The exact sample size (*n*) for each experimental group/condition, given as a discrete number and unit of measurement |
| ☒ | ☐ | A statement on whether measurements were taken from distinct samples or whether the same sample was measured repeatedly |
| ☒ | ☐ | The statistical test(s) used AND whether they are one- or two-sided <br> *Only common tests should be described solely by name; describe more complex techniques in the Methods section.* |
| ☒ | ☐ | A description of all covariates tested |
| ☒ | ☐ | A description of any assumptions or corrections, such as tests of normality and adjustment for multiple comparisons |
| ☒ | ☐ | A full description of the statistical parameters including central tendency (e.g. means) or other basic estimates (e.g. regression coefficient) AND variation (e.g. standard deviation) or associated estimates of uncertainty (e.g. confidence intervals) |
| ☐ | ☒ | For null hypothesis testing, the test statistic (e.g. *F*, *t*, *r*) with confidence intervals, effect sizes, degrees of freedom and *P* value noted <br> *Give P values as exact values whenever suitable.* |
| ☒ | ☐ | For Bayesian analysis, information on the choice of priors and Markov chain Monte Carlo settings |
| ☒ | ☐ | For hierarchical and complex designs, identification of the appropriate level for tests and full reporting of outcomes |
| ☒ | ☐ | Estimates of effect sizes (e.g. Cohen's *d*, Pearson's *r*), indicating how they were calculated |

*Our web collection on statistics for biologists contains articles on many of the points above.*

## Software and code

Policy information about availability of computer code

| Data collection | Spacedust is free open-source (GPLv3) software at: https://github.com/soedinglab/spacedust. <br><br> Software used as part of Spacedust: <br> MMseqs2 (15-6f452) :https://mmseqs.com/ <br> Foldseek (10-941cd33): https://github.com/steineggerlab/foldseek <br> ProstT5 (v0.0.1):https://github.com/mheinzinger/ProstT5 <br><br> Software used for annotating genes/proteins: <br> Prodigal (2.6.3): https://github.com/hyattpd/Prodigal <br> eggNOG-mapper (v2.0): https://github.com/eggnogdb/eggnog-mapper <br> AntiSMASH (v6.0.0): https://github.com/antismash/antismash <br><br> Software used for selection of genomes: <br> Mash (v2.3): https://github.com/marbl/Mash <br><br> Benchmarked software: <br> PADLOC (v1.1.0): https://github.com/padlocbio/padloc <br> ClusterFinder (Git: 5ee2c15): https://github.com/petercim/ClusterFinder <br> DeepBGC (v0.1.29): https://github.com/Merck/deepbgc <br> GECCO(v0.9.10):https://github.com/zellerlab/GECCO |
|---|---|

| Data analysis | Benchmark data and visualization was done with R/4.1.2, ggplot2/3.4.2, cowplot 1.1.1, gggenes/0.5.5, python/3.10.2, matplotlib/3.5.1 SciPy/1.8.0.<br>Scripts and data for data analysis are deposited in: https://wwwuser.gwdg.de/~compbiol/spacedust/ |
|---|---|

For manuscripts utilizing custom algorithms or software that are central to the research but not yet described in published literature, software must be made available to editors and reviewers. We strongly encourage code deposition in a community repository (e.g. GitHub). See the Nature Portfolio guidelines for submitting code & software for further information.

## Data

Policy information about availability of data

All manuscripts must include a data availability statement. This statement should provide the following information, where applicable:
- Accession codes, unique identifiers, or web links for publicly available datasets
- A description of any restrictions on data availability
- For clinical datasets or third party data, please ensure that the statement adheres to our policy

Data used in this work were obtained from the public sources and are freely accessible. Bacterial genome dataset was assembled from the KEGG GENOME collection (https://www.genome.jp/kegg/tables/br08606.html) downloaded from NCBI GenBank in 09/2022. The protein structure database used for mapping was compiled from the AlphaFold DB (https://alphafold.ebi.ac.uk/). The genomes and datasets used for the analysis in Fig. 5 and Supplementary Figs. 7–9 are publicly accessible via the supplementary material of Hannigan, G.D. et al. (https://doi.org/10.1093/nar/gkz654). The genome used as a query in Fig. 6 is available from NCBI GenBank under accession number NZ_BEXT01000001.1 (https://www.ncbi.nlm.nih.gov/nuccore/BEXT01000001.1). The target database is publicly available from GTDB (https://gtdb.ecogenomic.org/).
There are no restrictions on data availability, and all materials can be accessed freely.

## Human research participants

Policy information about studies involving human research participants and Sex and Gender in Research.

| Reporting on sex and gender | Not applicable |
|---|---|
| Population characteristics | Not applicable |
| Recruitment | Not applicable |
| Ethics oversight | Not applicable |

Note that full information on the approval of the study protocol must also be provided in the manuscript.

# Field-specific reporting

Please select the one below that is the best fit for your research. If you are not sure, read the appropriate sections before making your selection.

☒ Life sciences  ☐ Behavioural & social sciences  ☐ Ecological, evolutionary & environmental sciences

For a reference copy of the document with all sections, see nature.com/documents/nr-reporting-summary-flat.pdf

# Life sciences study design

All studies must disclose on these points even when the disclosure is negative.

| Sample size | Not applicable. Spacedust is an exclusively computational method and did not involve any experimental samples that would require sample size estimations, replications, blinding, or randomization. |
|---|---|
| Data exclusions | No data were excluded. |
| Replication | Not applicable. Spacedust is an exclusively computational method. The computational method is deterministic when run on the same computer setup. Therefore, replication is not needed as the result would be identical for each replicate. |
| Randomization | Not applicable. |
| Blinding | Not applicable. |

# Reporting for specific materials, systems and methods

We require information from authors about some types of materials, experimental systems and methods used in many studies. Here, indicate whether each material, system or method listed is relevant to your study. If you are not sure if a list item applies to your research, read the appropriate section before selecting a response.

## Materials & experimental systems

| n/a | Involved in the study |
|-----|----------------------|
| ☒ ☐ | Antibodies |
| ☒ ☐ | Eukaryotic cell lines |
| ☒ ☐ | Palaeontology and archaeology |
| ☒ ☐ | Animals and other organisms |
| ☒ ☐ | Clinical data |
| ☒ ☐ | Dual use research of concern |

## Methods

| n/a | Involved in the study |
|-----|----------------------|
| ☒ ☐ | ChIP-seq |
| ☒ ☐ | Flow cytometry |
| ☒ ☐ | MRI-based neuroimaging |

