## [Peer Review File · Nature Methods]

De novo discovery of conserved gene clusters in microbial genomes with Spacedust

Corresponding Author: Dr Johannes Soeding

Version 0:

Decision Letter:

13th Dec 2024

Dear Dr Soeding,

I sincerely apologize for the long review process, which has been delayed due to one of the reviewers not responding to our follow-up emails. Your Article, "De novo discovery of conserved gene clusters in microbial genomes with Spacedust", has now been seen by 2 reviewers. As you will see from their comments below, although the reviewers find your work of considerable potential interest, they have raised a number of concerns. We are interested in the possibility of publishing your paper in Nature Methods, but would like to consider your response to these concerns before we reach a final decision on publication.

We therefore invite you to revise your manuscript to address these concerns. We are committed to providing a fair and constructive peer-review process. Do not hesitate to contact us if there are specific requests from the reviewers that you believe are technically impossible or unlikely to yield a meaningful outcome.

Link Redacted

We hope to receive your revised paper within 12 weeks. If you cannot send it within this time, please let us know. In this event, we will still be happy to reconsider your paper at a later date so long as nothing similar has been accepted for publication at Nature Methods or published elsewhere.

OPEN SCIENCE REQUIREMENTS

REPORTING SUMMARY AND EDITORIAL POLICY CHECKLISTS

DATA AVAILABILITY

All novel DNA and RNA sequencing data, protein sequences, genetic polymorphisms, linked genotype and phenotype data, gene expression data, macromolecular structures, and proteomics data must be deposited in a publicly accessible database, and accession codes and associated hyperlinks must be provided in the "Data Availability" section.

CODE AVAILABILITY

Please include a "Code Availability" subsection in the Online Methods which details how your custom code is made available. Only in rare cases (where code is not central to the main conclusions of the paper) is the statement "available upon request" allowed (and reasons should be specified).

SUPPLEMENTARY PROTOCOL

To help facilitate reproducibility and uptake of your method, we ask you to prepare a step-by-step Supplementary Protocol for the method described in this paper. We [encourage authors to share their step-by-step experimental protocols](https://www.nature.com/nature-research/editorial-policies/reporting-standards#protocols) on a protocol sharing platform of their choice and report the protocol DOI in the reference list. Nature Portfolio's protocols.io is a free-to-use and open resource for protocols; protocols deposited onto protocols.io are citable and can be linked from the published article. More details can found at [protocols.io](https://www.protocols.io/help/publish-articles).

ORCID

Nature Methods is committed to improving transparency in authorship. As part of our efforts in this direction, we are now requesting that all authors identified as 'corresponding author' on published papers create and link their Open Researcher and

Contributor Identifier (ORCID) with their account on the Manuscript Tracking System (MTS), prior to acceptance. This applies to primary research papers only. ORCID helps the scientific community achieve unambiguous attribution of all scholarly contributions. You can create and link your ORCID from the home page of the MTS by clicking on 'Modify my Springer Nature account'. For more information please visit www.springernature.com/orcid.

Sincerely,
Lei

Lei Tang, Ph.D.
Senior Editor
Nature Methods

Reviewers' Comments:

Reviewer #2 (Remarks to the Author):

In their manuscript "De novo discovery of conserved gene clusters in microbial genomes with Spacedust", Zhang et al present a cluster discovery tool for microbial genomes.

The authors went for a structure-based comparison, which to my knowledge is a novel approach. Perhaps one of the reasons others have shied away from this approach previously is the challenge of getting good structure mappings for a large percentage of the microbial genomic dark matter of "conserved hypothetical proteins", as the authors themselves describe running into on page 6, where they had to fall back to MMseq2 sequence-based searches instead of structure-based comparison.

I'm not very convinced by the authors' decision to compare themselves against ClusterFinder and DeepBGC in terms of their ability to recover biosynthetic gene clusters (BGCs). ClusterFinder is a methodology that was built and tuned one and a half decades ago, with the limitation on data availability and computational resources that go along with that. The BGC mining field moved on from ClusterFinder many years ago, see DOI: 10.1007/s10295-018-2115-4 for a longer critique of the tool. DeepBGC was released in 2019 to address some of the most egregious false positive issues of ClusterFinder, but again I would say the field has largely moved on from there to better-performing tools. I do notice that the authors introduce a third BGC detection tool, but then only use that to annotate the biosynthetic roles in their detected BGCs, rather than as a comparison. Seeing how antiSMASH is pretty much the gold standard in BGC detection tools, it's hard to come away from this without a feeling that the reason this comparison wasn't done was that the authors' tool didn't perform better than the specialised tool. As the authors propose a more general purpose tool, I would not consider this a problem in general, but the claim on page 2 "... achieves better results in identifying 207 manually annotated BGCs than two specialized tools" is really tempered by the decision to pick the two worst-performing specialized tools. Even outside the rule-based antiSMASH, there are more-recently released better-performing ML-based BGC detection tools that, while less reliable than antiSMASH, vastly outperform DeepBGC. See e.g. GECCO (DOI: 10.1101/2021.05.03.442509v1). Realistically, a more relevant tool to compare against would be synteruptor (DOI: 10.1093/nargab/lqae069), as that also is an approach around going for the gene clusters first and worrying about the BGC function second, and also has seen a release in this decade.

The authors' overall conclusions are sound and backed up by data presented in the manuscript, their methods are described reasonably well. The authors highlight some of the current limitations of the tool, and point out a path to further improve in that area for future releases.

Reviewer #2 (Remarks on code availability):

The project's source code is available on GitHub under an OSI-approved open source license. While I did not get around to install and run spacedust on one of our big machines to evaluate the authors' stated claims on performance, I did clone and build the project according to the instructions in the readme file and am happy to report that this worked flawlessly, not a given for bioinformatics tools.

Reviewer #3 (Remarks to the Author):

Zhang et al. present a novel method to determine conserved gene clusters, which they named Spacedust. Overall, the manuscript is well written and describes a novel methodology which represents a conceptual advance over previous methods for this tool. Using structural conservation to detect remote homology is only possible for a short time and Spacedust is a welcome addition to the small array of tools already utilizing it.

Even though I support the publication of this manuscript, I have some comments.

Major comments:

Performance:

The dataset used for Spacedust is rather small, using only around 7100 genomes. It would be good if the authors could estimate how the performance of Spacedust scales with the number of input genomes. I don't expect the authors to use more genomes for this, reporting how long it would take for 2000 and 4000 genomes (for example) would give a good estimate already.

Benchmarking:

It would be very helpful to benchmark Spacedust using only the structure based approach (as well as only the homology based approach). The implemented mixed approach using mmseqs is pragmatic, but it would be good for users to have a better indication of the performance of the two different annotation methodologies underlying the final result.

Minor comments:

Page 2: It seems that spacedust is intrinsically reference dependent, even if an explicit reference database might not be necessary. It would be better to reformulate the sentence after (i)

Page 7: The section describing how limitation II is addressed is confusing. Maybe this paragraph needs to be rewritten for clarity. Hitchhiking and KEGG pathway annotation issues seem both relevant but distinct and are somehow combined here

A more modern installation procedure using a package manager or similar would be preferred

Reviewer #3 (Remarks on code availability):

The code seems professional. I cannot assess the cpp code though. I was able to install the tool via precompiled executables and it worked for me. The readme and other instructions are well written and easy to understand. It certainly is a usable resource for the community.

The code for reproducing the results of the paper is lacking a bit though. I downloaded the file https://wwwuser.gwdguser.de/~compbiol/spacedust/KEGG_70.tar.gz (the only file in the folder mentioned in the manuscript) and I could not figure out how to use the containing database to reproduce the manuscripts results.

Version 1:

Decision Letter:

Our ref: NMETH-A58101A

14th May 2025

Dear Dr. Soeding,

Thank you for submitting your revised manuscript "De novo discovery of conserved gene clusters in microbial genomes with Spacedust" (NMETH-A58101A). It has now been seen by the original referees and their comments are below. The reviewers find that the paper has improved in revision, and therefore we'll be happy in principle to publish it in Nature Methods, pending minor revisions to satisfy the referees' final requests and to comply with our editorial and formatting guidelines. As discussed in our previous email correspondence, we encourage you to include the comparison against the latest antiSMASH, if it doesn't take substantial efforts.

In the meantime, we are now performing detailed checks on your paper and will send you a checklist detailing our editorial and formatting requirements within two weeks or so. Please do not upload the final materials and make any revisions until you receive this additional information from us.

TRANSPARENT PEER REVIEW

Please note: we allow redactions to authors' rebuttal and reviewer comments in the interest of confidentiality. If you are concerned about the release of confidential data, please let us know specifically what information you would like to have removed. Please note that we cannot incorporate redactions for any other reasons. Reviewer names will be published in the peer review files if the reviewer signed the comments to authors, or if reviewers explicitly agree to release their name. For more information, please refer to our <https://www.nature.com/documents/nr-transparent-peer-review.pdf> target="new">FAQ page.

ORCID

IMPORTANT: Non-corresponding authors do not have to link their ORCID but are encouraged to do so. Please note that it will not be possible to add/modify ORCID at proof. Thus, please let your co-authors know that if they wish to have their ORCID added to the paper they must follow the procedure described in the following link prior to acceptance:
<https://www.springernature.com/gp/researchers/orcid/orcid-for-nature-research>

Sincerely,
Lei

Lei Tang, Ph.D.
Senior Editor
Nature Methods

Reviewer #2 (Remarks on code availability):

I did not re-review the code, but I remain happy with the quality of the install instructions and how easy it was to build the tool. The addition of conda packages and OCI containers makes it even easier to get spacedust installed.

Reviewer #3 (Remarks to the Author):

All my previous comments were addressed and I don't have additional comments. I support the publication of the manuscript.

The only comment I have pertains to the produced resource. It would be nice if the standard spacedust output file would be included (the .tsv file mentioned on the github page)

Reviewer #3 (Remarks on code availability):

Code works as before and the readme for the reproducibility is present now. Thanks!

Version 2:

Decision Letter:

13th Aug 2025

Dear Dr Soeding,

I am pleased to inform you that your Article, "De novo discovery of conserved gene clusters in microbial genomes with Spacedust", has now been accepted for publication in Nature Methods. The received and accepted dates will be 2nd Oct 2024 and 13th Aug 2025. This note is intended to let you know what to expect from us over the next month or so, and to let you know where to address any further questions.

Over the next few weeks, your paper will be copyedited to ensure that it conforms to Nature Methods style. Once your paper is typeset, you will receive an email with a link to choose the appropriate publishing options for your paper and our Author Services team will be in touch regarding any additional information that may be required. It is extremely important that you let us know now whether you will be difficult to contact over the next month. If this is the case, we ask that you send us the contact information (email, phone and fax) of someone who will be able to check the proofs and deal with any last-minute problems.

Authors may need to take specific actions to achieve compliance with funder and institutional open access mandates.

If your research is supported by a funder that requires immediate open access (e.g. according to [Plan S principles](https://www.springernature.com/gp/open-science/plan-s-compliance) or the [NIH public access policy](https://www.springernature.com/gp/open-science/us-federal-agency-compliance)) then you should select the gold OA route, and we will direct you to the compliant route where possible. Because authors warrant under our subscription licensing terms that they haven't committed to licensing any version of their article under a licence inconsistent with the terms of our agreement – including the applicable embargo period – publication under the subscription model isn't suitable for authors whose funders require no embargo.

Best regards,
Lei

Lei Tang, Ph.D.
Senior Editor
Nature Methods

** Visit the Springer Nature Editorial and Publishing website at http://editorial-jobs.springernature.com?utm_source=ejP_NMeth_email&utm_medium=ejP_NMeth_email&utm_campaign=ejp_Nmeth for more information about our career opportunities. If you have any questions please click [here](mailto:editorial.publishing.jobs@springernature.com).**

Reviewers' Comments:

Reviewer #2 (Remarks to the Author):

In their manuscript “De novo discovery of conserved gene clusters in microbial genomes with Spacedust”, Zhang et al present a cluster discovery tool for microbial genomes.

The authors went for a structure-based comparison, which to my knowledge is a novel approach. Perhaps one of the reasons others have shied away from this approach previously is the challenge of getting good structure mappings for a large percentage of the microbial genomic dark matter of “conserved hypothetical proteins”, as the authors themselves describe running into on page 6, where they had to fall back to MMseq2 sequence-based searches instead of structure-based comparison.

I’m not very convinced by the authors’ decision to compare themselves against ClusterFinder and DeepBGC in terms of their ability to recover biosynthetic gene clusters (BGCs). ClusterFinder is a methodology that was built and tuned one and a half decades ago, with the limitation on data availability and computational resources that go along with that. The BGC mining field moved on from ClusterFinder many years ago, see DOI: 10.1007/s10295-018-2115-4 for a longer critique of the tool. DeepBGC was released in 2019 to address some of the most egregious false positive issues of ClusterFinder, but again I would say the field has largely moved on from there to better-performing tools. I do notice that the authors introduce a third BGC detection tool, but then only use that to annotate the biosynthetic roles in their detected BGCs, rather than as a comparison. Seeing how antiSMASH is pretty much the gold standard in BGC detection tools, it’s hard to come away from this without a feeling that the reason this comparison wasn’t done was that the authors’ tool didn’t perform better than the specialised tool. As the authors propose a more general purpose tool, I would not consider this a problem in general, but the claim on page 2 “... achieves better results in identifying 207 manually annotated BGCs than two specialized tools” is really tempered by the decision to pick the two worst-performing specialized tools. Even outside the rule-based antiSMASH, there are more-recently released better-performing ML-based BGC detection tools that, while less reliable than antiSMASH, vastly outperform DeepBGC. See e.g. GECCO (DOI: 10.1101/2021.05.03.442509v1). Realistically, a more relevant tool to compare against would be synteruptor (DOI: 10.1093/nargab/lqae069), as that also is an approach around going for the gene clusters first and worrying about the BGC function second, and also has seen a release in this decade.

We thank the reviewer for pointing this out. We have now included the more recent ML-based tool GECCO suggested by the reviewer. We ran GECCO on the 9-genome gold standard dataset and included the result in Figure 5c-d and Supplemental Figures S7 and S8. GECCO performs slightly better than DeepBGC and similarly to ClusterFinder but not as well as Spacedust. It also tends to overpredict BGC regions. Synteruptor, the other tool mentioned by the reviewer, requires closely related genomes as input set and can only conduct all-against-all comparisons, whereas in this benchmark we would require the tool to compare the 9 genomes with a reference dataset.

The authors’ overall conclusions are sound and backed up by data presented in the manuscript, their methods are described reasonably well. The authors highlight some of the current limitations of the tool, and point out a path to further improve in that area for future releases.

Reviewer #2 (Remarks on code availability):

The project's source code is available on GitHub under an OSI-approved open source license. While I did not get around to install and run spacedust on one of our big machines to evaluate the authors' stated claims on performance, I did clone and build the project according to the instructions in the readme file and am happy to report that this worked flawlessly, not a given for bioinformatics tools.

We thank the reviewer for this assessment.

Reviewer #3 (Remarks to the Author):

Zhang et al. present a novel method to determine conserved gene clusters, which they named Spacedust. Overall, the manuscript is well written and describes a novel methodology which represents a conceptual advance over previous methods for this tool. Using structural conservation to detect remote homology is only possible for a short time and Spacedust is a welcome addition to the small array of tools already utilizing it. Even though I support the publication of this manuscript, I have some comments.

Major comments:

Performance:

The dataset used for Spacedust is rather small, using only around 7100 genomes. It would be good if the authors could estimate how the performance of Spacedust scales with the number of input genomes. I don't expect the authors to use more genomes for this, reporting how long it would take for 2000 and 4000 genomes (for example) would give a good estimate already.

Thank you! Spacedust has a quadratic runtime complexity with respect to the summed size of genomes, and this is now pointed out clearly.

In the Results:

“The run time of Spacedust scales quadratically with the summed coding length of genomes, due to the all-vs-all protein similarity search and cluster detection.”

In the discussion:

“Another limitation of Spacedust is its quadratic scaling of run time with the summed length of proteins in the genomes to be analyzed, caused by the quadratic time complexity of the all-versus-all comparison of proteomes and of the cluster detection algorithm.”

Benchmarking:

It would be very helpful to benchmark Spacedust using only the structure based approach (as well as only the homology based approach). The implemented mixed approach using mmseqs is pragmatic, but it would be good for users to have a better indication of the performance of the two different annotation methodologies underlying the final result.

Thank you for this comment. We have now extended Spacedust with the option to predict the 3Di sequence using the ProST5 model to obtain the structure sequences for all the proteins, without the need to map to AlphaFold DB and losing a fraction of the proteins. We have updated the workflow description in the Methods section and Fig 1 to highlight the option to use ProST5. In the software's README, we clarified how to create the database with ProST5 prediction.

We have rerun the benchmark on 1308 genomes with ProstT5 generated 3Di sequences (purely structure-based approach), and updated the result in Fig 2. We show that the purely structure-based approach yields more accurate KEGG Module predictions than the Foldseek + MMseqs approach, measured by the area under the precision-recall curve (PR-AUC). Additionally, we mentioned the current limitation of ProstT5 prediction in the discussion.

We also would like to point the reviewer to the Supplemental Figure S2-4, where we compared the heat map indicating protein matches with Foldseek + MMseqs, with only Foldseek (ProstT5) and only MMseqs.

Minor comments:

Page 2: It seems that spacedust is intrinsically reference dependent, even if an explicit reference database might not be necessary. It would be better to reformulate the sentence after (i).

We thank the reviewer for pointing this out. Because of the recent integration of ProstT5, Alphafold DB will not be required anymore as a reference database. EggNOGmapper still uses a reference database. We therefore reformulated the sentence under (i) in the introduction in the following way:

“[Spacedust] is reference-free and can discover conserved clusters of any type and composition; ...”
=>

“Spacedust discovers conserved gene clusters de novo, without relying on any prior information of the nature of gene clusters; ...”

Page 7: The section describing how limitation II is addressed is confusing. Maybe this paragraph needs to be rewritten for clarity. Hitchhiking and KEGG pathway annotation issues seem both relevant but distinct and are somehow combined here.

We agree that this paragraph was confusing. We reformulated it and omitted the last two sentences referring to possible negative bias of our KEGG-based analysis on Spacedust accuracy as this point is not so important and also quite speculative.

A more modern installation procedure using a package manager or similar would be preferred.

Spacedust can now be installed using a Docker container and Bioconda. We have also updated the corresponding installation instructions in the software's README file.

Reviewer #3 (Remarks on code availability):

The code seems professional. I cannot assess the cpp code though. I was able to install the tool via precompiled executables and it worked for me. The readme and other instructions are well written and easy to understand. It certainly is a usable resource for the community.

The code for reproducing the results of the paper is lacking a bit though. I downloaded the file https://wwwuser.gwdguser.de/~compbiol/spacedust/KEGG_70.tar.gz (the only file in the folder

mentioned in the manuscript) and I could not figure out how to use the containing database to reproduce the manuscripts results.

To facilitate reproducing the results of this study, we make the scripts for reproducing the all-vs-all search and visualizing the results available at <https://wwwuser.gwdguser.de/~compbiol/spacedust/>

"Data

Availability

...

The dataset and resource produced in this work are available at <https://wwwuser.gwdg.de/~compbiol/spacedust/>."

We would like to thank our reviewers for helping us to improve our work.

Ruoshi Zhang and Johannes Söding

Reviewers' Comments:

Reviewer #2 (Remarks to the Author):

The authors have addressed my concerns to my satisfaction. I still think that the main tool to compete with spacedust in the BGC detection arena is synteruptor, but I appreciate that the requirement of having closely related genomes is not compatible with the benchmark the authors are using. This is the main limitation to using synteruptor in day-to-day work as well, so I am excited to see another solution added to the mix. I am a bit surprised that GECCO didn't perform better in the benchmark, but the last time I ran a comparable benchmark was with an older GECCO version, so some things might have changed in the meantime.

While I was looking at the table of software versions in the SI, I noticed that the authors have been running antiSMASH 6 to get the functional annotations. With the antiSMASH authors just having released version 8, can the authors comment on why they used as old a version? Obviously I'm not expecting the authors to have used antiSMASH 8 for this, but was there a rationale for using version 6 instead of version 7 that the rest of us have been using in the last two years?

We reran the annotation with antiSMASH version 8 (instead of version 6) and updated Figure S10.

Reviewer #2 (Remarks on code availability):

I did not re-review the code, but I remain happy with the quality of the install instructions and how easy it was to build the tool. The addition of conda packages and OCI containers makes it even easier to get spacedust installed.

Reviewer #3 (Remarks to the Author):

All my previous comments were addressed and I don't have additional comments. I support the publication of the manuscript.

The only comment I have pertains to the produced resource. It would be nice if the standard spacedust output file would be included (the .tsv file mentioned on the github page)

Thank you for the idea. We added the file "results.tsv" to the github repository and uploaded it as supplementary file to Nature Methods.

Reviewer #3 (Remarks on code availability):

Code works as before and the readme for the reproducibility is present now. Thanks!

We would like to thank our reviewers for helping us to improve our work.

Ruoshi Zhang and Johannes Söding